# A systematic review and Bayesian meta-analysis provide evidence for an effect of acute physical activity on cognition in young adults
Jordan Garrett [1,2] ✉, Carly Chak[1,2], Tom Bullock[1,2] & Barry Giesbrecht [1,2] ✉

Physical exercise is a potential intervention for enhancing cognitive function across the lifespan. However, while studies employing long-term exercise interventions consistently show positive effects on cognition, studies using single acute bouts have produced mixed results. Here, a systematic review and meta-analysis was conducted to determine the impact of acute exercise on cognitive task performance in healthy young adults. A Bayesian hierarchical model quantified probabilistic evidence for a modulatory relationship by synthesizing 651 effect sizes from 113 studies from PsychInfo and Google Scholar representing 4,390 participants. Publication bias was mitigated using the trim-and-fill method. Acute exercise was found to have a small beneficial effect on cognition ($g = 0.13 \pm 0.04$; BF = 3.67) and decrease reaction time. A meta-analysis restricted to executive function tasks revealed improvements in working memory and inhibition. Meta-analytic estimates were consistent across multiple priors and likelihood functions. Physical activities were categorized based on exercise type (e.g., cycling) because many activities have aerobic and anaerobic components, but this approach may limit comparison to studies that categorize activities based on metabolic demands. The current study provides an updated synthesis of the existing literature and insights into the robustness of acute exercise-induced effects on cognition. Funding provided by the United States Army Research Office.

A single bout of exercise induces a cascade of neuromodulatory changes that influence multiple brain systems[1,2]. This includes an increase in the synthesis of neurotransmitters (e.g., acetylcholine, dopamine, GABA, glutamate) and neurotrophic factors (e.g., BDNF), which can occur in a brain-region-specific manner (see ref. 1 for review). Given these impacts on the brain, it would be reasonable to hypothesize that single brief bouts of exercise are associated with changes in performance across a range of cognitive domains. Consistent with this hypothesis, there is abundant evidence that attention[3–6], working memory[7–11], decision making[12,13], and cognitive control[14,15] are facilitated by brief bouts of physical exercise. However, there is also evidence suggesting that exercise has little or no effect on cognitive task performance. For instance, Komiyama et al.[16] observed no difference in accuracy on a spatial delayed response task between exercise and rest conditions. Further, working memory performance has been shown to remain unchanged either during or after a single bout of exercise[6]. The discrepant pattern of results in the literature investigating the link between exercise and performance on

cognitive tasks is surprising given the consistent and robust physiological effects of even brief bouts of physical activity. However, it is unclear whether this limited impact of exercise on performance reflects the true state of affairs or whether the apparent lack of robust influence is due to vast empirical discrepancies across studies in the literature. Studying the impact of single exercise sessions on cognition can provide insight into how changes in our body's physiological state impact behavior. This understanding can then guide the creation of more effective longer-term exercise interventions, which essentially involve regularly repeating brief exercise sessions over an extended period.

Meta-analytic techniques are a set of powerful tools that can expose dominant trends within a methodologically heterogeneous literature. There is a consensus amongst narrative reviews and previous meta-analyses that an acute bout of exercise has a small positive influence on behavioral performance[1,17–23]. The nature of this effect is moderated by exercise protocol, cognitive tasks, and participant characteristics. For instance,

[1]Department of Psychological & Brain Sciences, University of California, Santa Barbara, CA, USA. [2]Institute for Collaborative Biotechnologies, University of California, Santa Barbara, CA, USA. ✉e-mail: jordangarrett@ucsb.edu; giesbrecht@ucsb.edu

Lambourne & Tomporowski[20] observed that task performance during exercise was dependent on exercise modality, the type of cognitive task, and when it was completed relative to exercise onset. Similarly, post-exercise performance was moderated by exercise modality and the type of cognitive task. Chang et al.[18] reported that post-exercise cognitive performance was influenced by exercise intensity, duration, and the time of cognitive test relative to exercise cessation. Interestingly, the authors found that study sample age was a significant moderator, where larger positive effects were found for high school (14–17 years), adult (31–60 years), and older adult (>60 years) samples compared to elementary (6–13 years) and young adult (18–30 years) samples. Multiple meta-analyses have observed that the effect of exercise is dependent on cognitive domain, with measures of executive function, attention, crystallized intelligence, and information processing speed showing the largest gains[18,19,24–26]. Further, there is evidence that exercise has a differential influence on the speed and accuracy of cognitive processes. McMorris et al.[21] observed that acute, intermediate exercise facilitated response times on working memory tasks, while accuracy was compromised. In contrast, exercise has been shown to boost both the accuracy and speed of cognitive control[23]. Altogether, it is important to consider cognitive task, participant, and physical activity characteristics to develop a holistic model of the relationship between exercise and cognition.

While these earlier meta-analyses have provided unique insights into understanding the relationship between acute exercise and cognition, they have two major limitations. First, the most recent holistic quantitative synthesis of the extant literature was published over a decade ago[18]. Meanwhile, the exercise and cognition literature has grown drastically. According to the electronic database Web of Science, almost 6,000 articles associated with the search term "exercise and cognition" have been published since this last holistic meta-analysis. In addition, more recent meta-analyses have primarily focused on executive processes[19,22,26,27]. Thus, previous models may provide an outdated and limited account of exercise-induced influences on other aspects of cognition, such as perception, long-term memory, and learning. Second, previous meta-analytic approaches employed frequentist statistical methods, which are based on a decision threshold rather than a characterization of the relevant evidence. As a result, it is possible that acute exercise and moderator variables are deemed to have a significant influence on task performance despite the fact that there may only be a small degree of probabilistic evidence in favor of this notion. In addition, relying on a decision threshold prevents these models from conveying the likelihood that an exercise protocol elicits a change in cognitive task performance. Past frequentist meta-analytic models also treated heterogeneity parameters as a fixed quantity and utilize only a point estimate, which can lead to an underestimation of the variability either between or within studies[28–31]. This is especially true when the number of modeled studies is low[32–34]. When considered together, there is a clear need for an updated meta-analysis using an approach that addresses these limitations.

The current study addressed these limitations in two ways. First, a comprehensive literature search was conducted spanning the years 1995–2023. To quantify the influence of exercise on cognition in young healthy adults, the search was limited to non-clinical studies whose subjects were between 18–45 years old. The analysis focused on subjects within this age range since exercise research has predominantly been dedicated toward studying the effects in children and older adults[35,36]. Studies were required to be experimental in nature, and consist of both an acute exercise manipulation and cognitive task measurements. A broad range of cognitive domains encompassing tasks probing perception to executive function were included in the meta-analysis. Similarly, a wide range of exercise types and testing contexts were included. For example, traditional laboratory exposures to exercise (e.g., cycling, running) and sport activities in real-world settings were viable candidates for analysis. By casting a wide net, the current study provides a large scope and updated summary of the current state of the exercise and cognition literature.

Second, the current study uses a Bayesian meta-analytic approach to synthesize studies across the acute exercise and cognition literature. The Bayesian approach affords a flexible modeling framework that uses reported effect sizes to characterize the relative evidence in favor of a modulatory account. Inherently, a random effects meta-analytic model is hierarchical in nature, making it well suited for Bayesian methods. When utilized within this statistical framework, priors are placed on parameters at the highest level of the model such as the estimated pooled effect size and measures of heterogeneity. This approach has several advantages compared to its frequentist counterpart. First, the use of priors on heterogeneity parameters can attenuate the underestimation of variation both between and within studies[37,38], leading to a clearer understanding of sources of heterogeneity and an increased precision when estimating the pooled effect size[39]. Furthermore, priors provide additional constraints on low-level parameter estimates and a greater degree of "shrinkage" of outliers towards the overall pooled effect size or mode(s) of grouping variables[39,40]. Therefore, a Bayesian meta-analysis is more robust to outliers and can be more conservative when proper priors are employed. Second, the method yields a posterior distribution for all parameter estimates. This grants the capability of directly modeling the degree of uncertainty in heterogeneity estimates[37]. Posterior distributions can be used to compute the probability that an exercise protocol elicits a change in task performance of a given magnitude (e.g., large effect size). Compared to the approximation of $p$-values and confidence intervals, which require additional assumptions for hierarchical models, calculating the high-density interval (HDI), which indicates the most credible outcomes in the posterior distribution, for complex hierarchical models is seamless[39]. Third, it is possible to incorporate knowledge from previous meta-analyses when constructing prior distributions. This affords the ability to quantitatively compare the observed data to the predictions of previous models.

Considering the results of past meta-analyses, exercise was expected to have a small positive influence on cognition. Cognitive task and exercise characteristics were anticipated to moderate this relationship, as evidenced by nonzero parameter estimates, reflecting the selective nature of exercise-induced effects. Model comparisons were conducted to evaluate how moderator inclusion improved predictive performance, and robustness of parameter estimates were determined by employing multiple priors and likelihood functions.

## Methods

### Literature search
Studies investigating the impact of an acute bout of exercise on cognition were obtained through searches of the electronic databases PsychInfo and Google Scholar according to the PRISMA guidelines[41]. On 09 September 2023, databases were queried using a search string that combined the following physical activity and cognitive domain keywords: ["exercise" OR "physical activity" OR "physical exertion" OR "physical fatigue"] AND ["perception" OR "attention" OR "working memory" OR "executive function" OR "memory" OR "decision making" OR "motor skill" OR "skill acquisition" OR "language" OR "reasoning"]. For the PsychInfo search, the filters "journal article", "English", "empirical study", "human", and "peer reviewed" were applied. Search results were limited to studies published between 1995 and 2023 and whose subjects were between 18 and 45 years of age. Note, this literature search and analysis were not preregistered, nor was a review protocol prepared prior to the literature search.

### Eligibility criteria
Studies were deemed eligible for inclusion in the meta-analysis if they met all of the following criteria: assessed the influence of an acute bout of exercise on cognition, compared the effects of exercise with an active and/or passive control group(s), utilized cognitive tasks that measured reaction time (RT) and/or accuracy, tested cognition either during, pre-, or post-exercise and consisted of cognitively normal subjects. Note, an acute bout was defined as an instance of physical activity that occurred within a single 24-hour period[18]. Two researchers independently screened records based on their title, abstract, and full text. In the case of discrepancies, a third researcher resolved them by reading the full-text.

## Data extraction and coding

Information concerning experimental design and procedures, exercise details (i.e., type, intensity, duration), and sample characteristics were extracted from the final list of studies by a single researcher. Means and standard deviations of accuracy and/or RT measures on all cognitive tasks were inserted into an electronic spreadsheet for the calculation of effect sizes. The primary outcome measures for each domain were inserted separately if a task assessed multiple cognitive domains. Regarding studies that probed cognition at multiple time points during or post-exercise, measures for each time point were also recorded separately. If the statistics necessary for calculating effect sizes were not reported in the full-text of the article, the authors were contacted and asked to provide them.

All effect sizes were categorized into one of seven cognitive domains that were generally based on the DSM-5[42]: executive function, information processing, perception, attention, learning, motor skills, and memory. The classification criteria used for categorizing a cognitive task into a domain is provided in the Supplementary Table 1. To account for variability in the metric used to measure exercise intensity across studies (e.g., ventilatory threshold, heart rate), each intensity was labeled as either light, moderate, or vigorous according to the American College of Sports Medicine guidelines[43]. Exercise durations were grouped into one of five time bins: ≤16 minutes, 20–27 minutes, 30–35 minutes, 40–45 minutes, ≥60 minutes. In the event that a study did not provide the exercise duration, its time bin was labeled as "not provided". Exercise types were based on the modality reported in each study, yielding the following categorizations: cycling, high intensity interval training (HIIT), running, walking, circuit training, resistance exercise, and sports activity. The latter category encompassed studies that used sports-related exercises that did not fit into the other labels, such as rock climbing or soccer. The time at which cognitive task performance was evaluated relative to exercise was categorized as either during exercise or 0, 15, 20–75, and ≥180 minutes after cessation. Lastly, effect sizes were also coded according to task performance dependent measures (i.e., RT vs accuracy). Note, the levels of each categorical moderator were chosen with the intention of achieving a balance between specificity and statistical power to yield reliable estimates that can inform the design of future exercise studies.

## Calculating effect sizes

Cohen's $d$ effect sizes were calculated for studies that tested cognition pre-/post-exercise without a control condition by dividing the mean change in performance by the standard deviation of the pre-test. If the study included a control group (e.g., rest), the mean change of the control condition was subtracted from the mean change of the exercise condition and divided by the pooled standard deviation of pretest scores[20,44]. For studies that tested cognition during, or only after exercise, the mean of the control condition was subtracted from the mean of the exercise condition and divided by the standard deviation of the control condition[21]. All effect sizes were converted into the bias-corrected standardized mean difference, Hedge's $g$, by multiplying them by the correction factor $J = 1 - \frac{3}{4df-1}$ where $df$ is the degrees of freedom[45]. The sign of effect sizes for RT and error were reversed to reflect a positive influence of exercise on cognitive task performance. Once effect sizes were extracted from each study, inspection of a funnel plot and Egger's regression test were conducted to assess the risk of publication bias.

## Bayesian hierarchical modeling

The overall effect of exercise on cognition was assessed using a Bayesian hierarchical model[46,47], which was implemented through the R package *brms*[48]. In the first level of the model, a study's observed effect size(s) $\hat{\theta}_{ik}$ was assumed to be an estimate of the true effect size $\theta_k$. The observed effect(s) $\hat{\theta}_{ik}$ were modeled as being sampled from a normally distributed population underlying study $k$ with a mean equivalent to the true effect and a variance of $\sigma_k^2$. In the second level of the model, the true effect size $\theta_k$ was assumed to have been drawn from an overarching distribution whose mean represented the overall pooled effect μ, and whose variance depicted the degree of between-study heterogeneity $\tau^2$. The final level of the model contained weakly informative priors. A standard normal prior was used for the pooled

effect, while the prior for $\tau^2$ was a Half-Cauchy distribution with location and scale parameters set to 0 and 0.5, respectively.

Following the main meta-analysis, subgroup analyses were conducted to determine potential moderators of the relationship between exercise and cognitive task performance. More specifically, we analyzed the influence of the following primary moderators: cognitive domain, time of cognitive test relative to exercise, task outcome measure, exercise intensity, duration, and type. The following secondary moderators were also analyzed to determine the influence of study and participant characteristics on the overall pooled effect size: average sample age, body mass index (BMI kg/m²), height (cm), weight (kg), VO2 max (ml/kg/min), percentage of female participants, within- vs between-study design, and publication year. With the exception of publication year and the percentage of female participants, all secondary moderators were mean centered for interpretability. A standard normal distribution was used as a weakly informative prior for the difference in effect sizes between subgroups. When reporting model parameter estimates, we use the [mode ± standard deviation] and the 89% HDI of posterior distribution.

## Statistical inference

For all estimated effect sizes, Bayes Factors (BFs) were used to determine the degree of evidence in favor of a difference from zero. BFs were approximated using the reciprocal of the Savage-Dickey density ratio, which was implemented using the function *bayesfactor_parameters* from the *bayestestR* package[49]. This method involves dividing the height of the prior distribution for the null value by the height of the posterior distribution at the same value, and represents the credibility of the null value for a parameter once the data has been taken into consideration. BFs were also used to ascertain the predictive performance of subgroup models. After each model was compared to a null counterpart (i.e., moderator excluded) using the function *bayesfactor_models*, an inclusion BF (*bayesfactor_inclusion*) was estimated to determine if including a moderator improved predictive power[50]. To estimate stable BFs, a large number of sampling iterations (10,000) and warmup samples (2000) were used for each of four chains when estimating model parameters[51]. BFs were interpreted following the guidelines proposed by Jeffreys [52]. A BF between 1 and 3 indicates "anecdotal" evidence for the alternative hypothesis, between 3 and 10 indicates "moderate" evidence, between 10 and 30 indicates "strong" evidence, and greater than 30 indicates "very strong" evidence[39,53–55]. The reciprocal of these ranges signifies evidence in favor of the null hypothesis (e.g., 0.33-1 = anecdotal evidence). When conducting subgroup analyses with more than two factors, orthonormal coding was employed to ensure that an identical prior was used for each factor level and that estimated BFs were accurate[56]. Parameter estimates were extracted from all models using the R package *emmeans*.

## Sensitivity analysis

A popular criticism of the Bayesian approach is that priors are chosen subjectively, which in turn can bias parameter estimates and their corresponding BFs[40,57]. Although utilizing weakly informative priors mitigates bias, a sensitivity analysis that evaluates the contribution of both priors and the likelihood function must be conducted to determine if the model results are robust[38,58–60]. Thus, we replicated the previously described modeling approach with the exception of using two different priors for the overall pooled effect size. The first was a normal distribution with a mean of zero and standard deviation of ½. Since this prior adds greater weight to the probability that exercise has no influence on task performance, we denoted it as the no effect (NE) prior. The second prior was constructed by synthesizing estimates from previous meta-analyses on acute exercise and cognition[18,20–23], resulting in a normal distribution with a mean of 0.24 and standard deviation of 0.57. This prior was denoted as the positive effect (PE) prior.

The influence of the likelihood function was assessed by modeling study effect sizes as being sampled from a *t*-distribution. An advantage of using this likelihood function, compared to a normal distribution, is that

**Fig. 1 | PRISMA flow diagram of literature search results.** A total of 113 studies were deemed eligible for meta-analytic modeling.

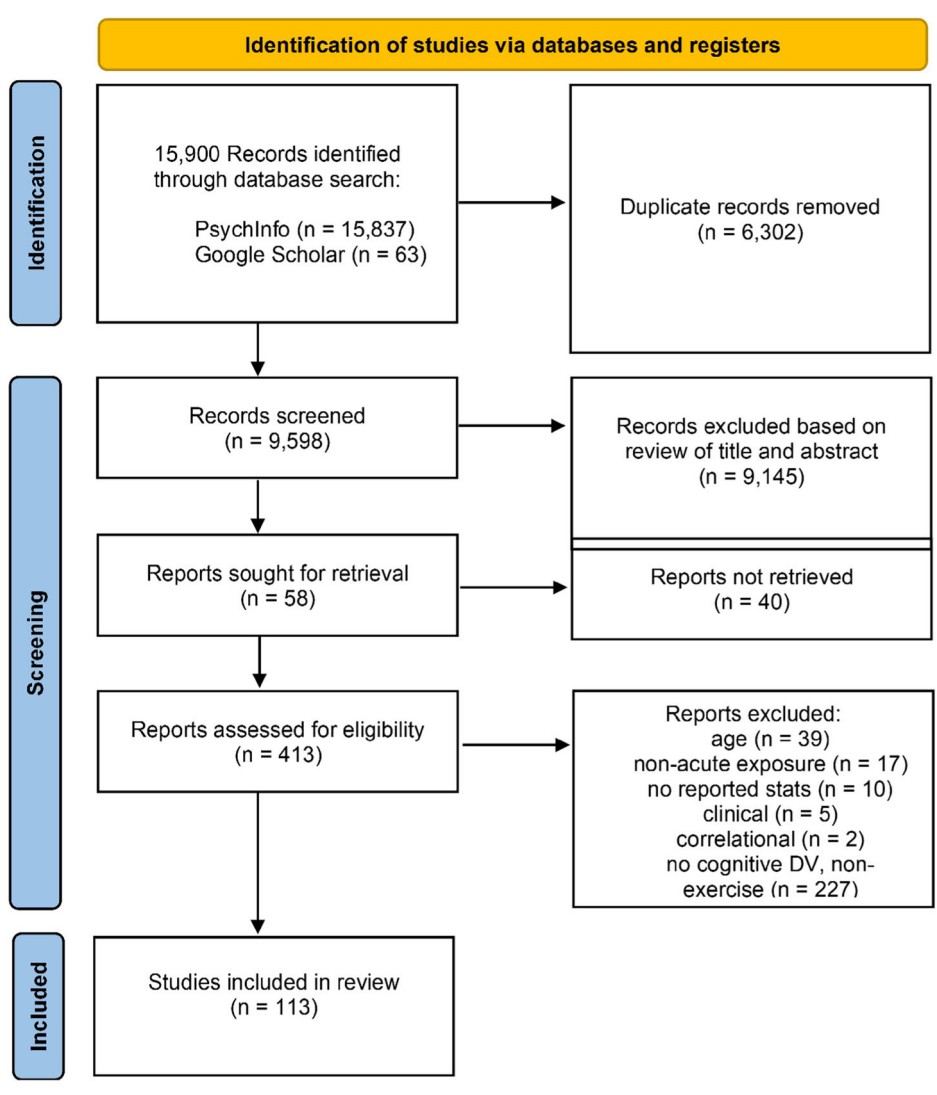

## Results

### Description of studies

The literature search yielded 15,900 peer reviewed journal articles, and after removing duplicates 8295 remained. Subsequent an initial screening based off the titles and abstracts, 805 studies were identified as potential candidates for modeling. 113 of these studies were deemed eligible for inclusion in the meta-analysis according to their full-text contents (Fig. 1). In total, 642 effect sizes were extracted from these studies, representing data from 4390 subjects. A majority of the effects measured the influence of exercise on executive function ($k = 434$) and attention ($k = 109$). Fewer effects were measured during exercise ($k = 82$) relative to after the cessation of exercise ($k = 560$). Visual inspection of a funnel plot suggested that the effect sizes were distributed symmetrically (Fig. 2a), however there was very strong evidence for asymmetry according to Egger's regression intercept ($\beta = 1.18 \pm 0.25$; HDI = [0.78, 1.58]; BF = 253.24) suggesting the presence of publication bias. This was addressed by employing the trim and fill

model parameter estimates are influenced less by outliers[40]. The Half-Cauchy prior was used for the scale of the distribution, while a standard normal prior was used for its mean. For its shape (i.e., degree of freedom) an exponential distribution with a rate equal to 1/29 served as a prior. To determine if meta-analytic estimates were robust across the alternative priors and likelihood function, we visually compared the posterior distributions across models for large deviations[58].

approach, which imputes low-precision effect sizes until the funnel plot is symmetrical[61].

### Overall effect

The meta-analysis indicated that there was moderate evidence for an acute bout of exercise to have a small positive influence on overall performance across cognitive domains ($g = 0.13 \pm 0.04$; HDI = [0.06, 0.20]; BF = 3.67) (Fig. 2b, d). According to the posterior distribution, there was a low probability that the estimated pooled effect was less than or equal to zero ($p = 0.01$) and an 80% chance that the effect size fell between the range of 0 to 0.2 (Fig. 2c). There was a large amount of heterogeneity within ($\tau_{within} = 0.65 \pm 0.03$; HDI = [0.60, 0.70]; $I^2_{within} = 81.19\%$) and moderate amount between ($\tau_{between} = 0.29 \pm 0.05$; HDI = [0.20, 0.38]; $I^2_{between} = 15.9\%$) studies (Fig. 2e). Effect size estimates for each individual study are presented in Table 1.

### Subgroup analyses

Primary subgroup analyses revealed that acute exercise reduced RT on cognitive tasks ($g = 0.27$; HDI = [0.18, 0.36]; BF= $6.71 \times 10^3$), but had no impact on accuracy ($g = 0.04$; HDI = [−0.04, 0.12]; BF = $6.15 \times 10^{-2}$) (Table 2) (Fig. 3a). Engaging in either cycling ($g = 0.21$; HDI = [0.11, 0.32]; BF = 14.74) or HIIT ($g = 0.73$; HDI = [0.40, 1.09]; BF = 26.05) was found to have an enhancing effect on performance in cognitive tasks (Fig. 3b). In regard to cognitive domain, there was evidence that acute exercise has a positive influence on executive processes ($g = 0.18$; HDI = [0.10, 0.27];

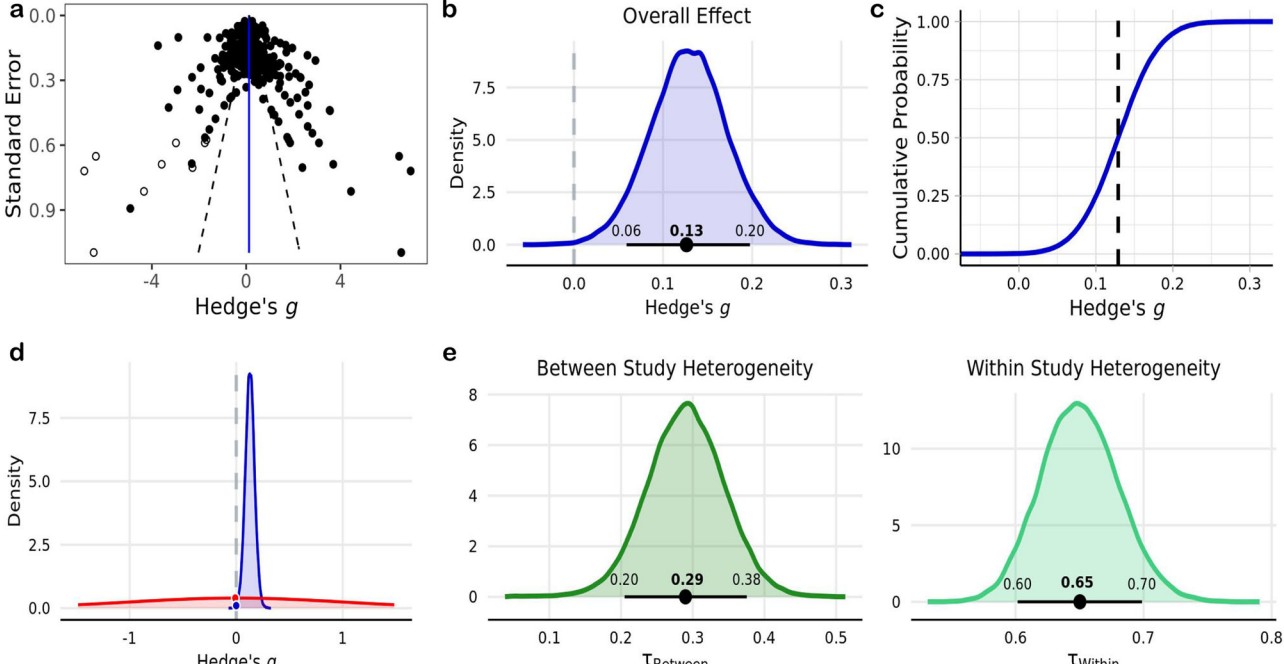

**Fig. 2 | Meta-analysis of the effect of acute exercise on general cognitive task performance. a** Funnel plot of 642 study effect sizes (black circles). Imputed effect sizes after using the trim and fill method are represented by the unfilled circles (n = 9). Vertical blue line indicates the estimated pooled effect sizes, while dashed black lines represent a pseudo 95% confidence limits. **b** Posterior distribution of estimated pooled effect. Horizontal black line indicates bounds of 89% HDI derived using n = 651 effect sizes. **c** Empirical cumulative density function of distribution in **b**, where the dashed black line indicates the pooled effect. **d** Representation of using the Savage-Dickey ratio to calculate BFs. The density of the null value in the prior distribution (red) is divided by its density in the posterior distribution (blue) to yield probabilistic evidence in favor of the alternative hypothesis. **e** Posterior distributions of between and within study heterogeneity.

BF = 36.97). Furthermore, behavioral performance was found to improve immediately after exercise cessation (g = 0.16; HDI = [0.11, 0.30]; BF = 4.03) and in response to vigorous intensity exercises (g = 0.19; HDI = [0.09, 0.28]; BF = 5.03). Lastly, at least moderate evidence in favor of non-zero parameter estimates were observed for the secondary moderators publication year, within-subjects design, age, percentage of female participants, and weight (Table 3).

To test for the possible contribution of a learning effect to the estimated overall pooled effect size, a separate meta-analysis was conducted on effects from studies employing a pre-/post-test design (N effect sizes = 298). Despite the estimated pooled effect size for this subset of data being nominally similar to the estimate for the entire dataset, there was anecdotal evidence in favor of the null hypothesis (g = 0.15 ± 0.06; HDI = [0.04, 0.24]; BF = 0.95). Moderator analyses indicated that there was no credible evidence for a difference in this estimated pooled effect size as a function of whether or not a control group was included in the study (BF_{Inclusion} = 0.12; w/control : g = 0.18 ± 0.10; HDI = [0.03, 0.33]; BF = 0.51; w/o control : g = 0.11 ± 0.13; HDI = [−0.03, 0.26]; BF = 0.18), suggesting that the estimated influence of exercise on general cognitive performance is not driven by a learning effect.

### Model comparisons

Model comparisons were performed to determine if including a moderator improved predictive performance. Only a model that included task performance measure as a moderator was more likely when compared to a null counterpart (BF_{Inclusion} = 357.10) (Table 4). This is likely due to a number of factors. First, acute exercise had a negligible impact on a majority of the levels in each subgroup. Second, there was a high degree of uncertainty in estimated model coefficients, as evidenced by their wide HDI intervals. Third, Bayesian inference automatically penalizes model complexity and favors more parsimonious models. If a model has many parameters, but a majority of them are nonzero, then a simpler counterpart will be favored.

### Interactions between moderators

An exploratory analysis was conducted to determine if the influence of moderator variables was contingent on one another. Due to the computationally intensive nature of Bayesian modeling, analyses were limited to the following pairs of moderators: (1) exercise intensity and type, (2) exercise intensity and duration, (3) exercise type and duration, (4) cognitive domain and exercise type, (5) cognitive domain and exercise intensity, (6) cognitive domain and task performance measure, (7) exercise type and task performance measure. Although none of the pairs of interaction models had more predictive power compared to a null counterpart (BF_{Inclusion}: Model 1 = 3.86 × 10^{−6}; Model 2 = 1.66 × 10^{−8}; Model 3 = 1.84 × 10^{−3}; Model 4 = 1.31 × 10^{−4}; Model 5 = 3.91 × 10^{−5}; Model 6 = 7.1 × 10^{−3}; Model 7 = 7.05 × 10^{−4}), there were two that had nonzero parameter estimates.

The first model included an interaction between cognitive domain and exercise type. There was evidence in favor of cycling improving performance on tasks that probed attention (g = 0.34; HDI = [0.14, 0.56]; BF = 3.05) and executive function (g = 0.28; HDI = [0.14, 0.40]; BF = 17.83). HIIT exercises were found to bolster executive function (g = 1.01; HDI = [0.61, 1.43]; BF = 155.33), while resistance exercises had an aversive impact on attentional performance (g = −0.76; HDI = [−1.20, −0.38]; BF = 18.07) (Fig. 4a). The second model included an interaction between cognitive domain and task performance measure and indicated that time-dependent measures of executive function are improved (g = 0.30; HDI = [0.19, 0.39]; BF = 1.10 × 10^3) (Fig. 4b).

### Sensitivity analyses

The estimated overall effect of acute exercise on cognition was consistent across the NE prior (g = 0.13 ± 0.04; HDI = [0.06, 0.20]; BF = 6.52), PE prior (g = 0.12 ± 0.04; HDI = [0.06, 0.19]; BF = 6.51), and t likelihood function (g = 0.12 ± 0.04; HDI = [0.06, 0.18]; BF = 8.77) (Fig. 5a). Interestingly, there was anecdotal-to-moderate evidence in favor of the synthesized estimate from previous meta-analyses (i.e., g = 0.24) across the PE (BF = 3.19), NE

**Table 1 | Study description and estimated effect sizes**

| Study | N | Experimental design | Exercise type | Exercise intensity | Duration (min) | Cognitive domain | Test time | N effects | g | 89% HDI |
|---|---|---|---|---|---|---|---|---|---|---|
| Hogervorst et al., [124] | 15 | Within | Cycling | Vigorous | 60 | Inhibition | Pre-Post | 1 | 0.03 | −0.39, 0.47 |
| McMorris & Graydon, 1996 | 20 | Within | Cycling | Moderate vigorous | 2 | Executive function | During | 6 | 0.11 | −0.19, 0.45 |
| Brisswalter et al., 1997 | 20 | Within | Cycling | Vigorous | 40 | Information processing | Pre-Post | 4 | 0 | −0.36, 0.35 |
| Arcelin & Delignieres, 1998 | 22 | Within | Cycling | Moderate | 10 | Information processing | During | 2 | 0.03 | −0.35, 0.44 |
| McMorris et al., 1999 | 9 | Within | Cycling | Moderate vigorous | Task completion | Executive function | During | 4 | 0.01 | −0.34, 0.38 |
| Collardeau et al., 2001 | 11 | Within | Running | Vigorous | 90 | Perception | Pre-Post | 2 | −0.05 | −0.47, 0.34 |
| Deuster et al., 2002 | 13 | Within | Running | Vigorous | Volitional Exhaustion | Attention Learning Executive function | Pre-Post | 24 | 0.16 | −0.05, 0.39 |
| Hillman et al., 2003 | 19 | Within | Running | Vigorous | 30 | Executive function | Post | 2 | −0.08 | −0.44, 0.35 |
| Pesce et al., [125] | 16 | Within | Cycling | Moderate | Task completion | Attention | During | 2 | 0.25 | −0.15, 0.70 |
| Davranche et al., 2005a | 7 | Within | Cycling | Maximal | Volitional exhaustion | Perception | Pre-Post | 2 | 0.08 | −0.32, 0.49 |
| Davranche & Pichon, 2005b | 12 | Within | Cycling | Maximal | Volitional exhaustion | Perception | During | 1 | 0.06 | −0.37, 0.49 |
| Pesce et al., [126] | 48 | Within | Cycling | Vigorous | Task duration | Attention | During | 2 | 0.14 | −0.24, 0.57 |
| Vickers & Williams, [127] | 10 | Within | Cycling | Moderate Vigorous | 2 | Motor Skills | Pre-Post | 4 | −0.05 | −0.43, 0.29 |
| Lo Bue-Estes et al., [128] | 26 | Between | Running | Vigorous | 20 | Executive function Memory | Pre-Post | 8 | 0.01 | −0.28, 0.32 |
| Coles & Tomporowski, [129] | 18 | Within | Cycling | Moderate | 40 | Executive function | Pre-Post | 1 | 0 | −0.41, 0.44 |
| Fontana et al., [130] | 32 | Within | Running | Light Moderate Vigorous | 4 | Executive function | During | 6 | 0.07 | −0.23, 0.40 |
| Luft et al., [131] | 30 | Within | Running | Vigorous | 20 | Attention information processing Memory | Pre-Post | 10 | −0.11 | −0.39, 0.16 |
| Pontifex et al., [132] | 21 | Within | Running Resistance | Vigorous | 30 | Executive function | Pre-Post | 4 | −0.31 | −0.69, 0.05 |
| Srygley et al., [133] | 52 | Within | Walking | Light | Task Completion | Executive function | During | 2 | 0.03 | −0.4, 0.39 |
| Thomson et al., [134] | 163 | Within | Treadmill | Vigorous | 27 | Executive function | Pre-Post | 2 | 0.1 | −0.3, 0.49 |
| Lambourne et al., [6] | 19 | Within | Cycling | Moderate | 40 | Executive function | Pre, During, Post | 4 | −0.36 | −0.76, 0 |
| Norling et al., [135] | 121 | Between | Running | Light Moderate Vigorous | 30 | Attention | Pre-Post | 3 | 0.08 | −0.28, 0.46 |
| Young et al., [136] | 27 | Within | Running | Moderate | 7 | Motor skills | Pre-Post | 1 | −0.02 | −0.47, 0.39 |
| Chang et al., [137] | 42 | Between | Cycling | Vigorous | 30 | Executive function | Pre-Post | 3 | −0.02 | −0.37, 0.35 |
| Green & Helton, [138] | 12 | Within | Climbing | Moderate | 3 | Memory | During | 1 | −0.2 | −0.71, 0.20 |
| Ohlinger et al., [139] | 50 | Within | Walking | Light | Task completion | Attention Executive function | During | 3 | −0.01 | −0.38, 0.35 |
| Hope et al., [140] | 52 | Between | Punching | Vigorous | Volitional exhaustion | Memory | Post | 3 | −0.42 | −0.83, −0.02 |
| Lambourne, [141] | 16 | Within | Cycling | Vigorous | 35 | Executive function | Pre-Post | 1 | −0.01 | −0.46, 0.39 |

**Table 1 (continued) | Study description and estimated effect sizes**

| Study | N | Experimental design | Exercise type | Exercise intensity | Duration (min) | Cognitive domain | Test time | N effects | g | 89% HDI |
|---|---|---|---|---|---|---|---|---|---|---|
| Moore et al., [142] | 30 | Between | Cycling | Moderate | 6–54 | Perception / Executive function / Attention | Pre-Post | 5 | −0.12 | −0.43, 0.23 |
| Quelhas Martins et al., [9] | 24 | Between | Cycling | Moderate | 8 | Executive function | During | 1 | 0.1 | −0.33, 0.53 |
| Roberts & Cole, [143] | 40 | Within | Circuit | Light Moderate | 1–5 | Executive function | Pre-Post | 8 | −0.05 | −0.34, 0.25 |
| Bullock & Giesbrecht, [144] | 26 | Between | Cycling | Moderate | 136 | Perception | Pre, Post | 8 | −0.14 | −0.44, 0.15 |
| Byun et al., [145] | 25 | Within | Cycling | Light | 10 | Executive function | Pre-Post | 2 | 0.17 | −0.23, 0.58 |
| Darling & Helton, [146] | 12 | Within | Climbing | Moderate | 5 | Memory | During | 2 | −0.1 | −0.51, 0.29 |
| Nibbeling et al., [147] | 22 | Between | Running | Vigorous | 10 | Motor Skills / Executive function / Memory / Attention | Post | 5 | −0.08 | −0.43, 0.24 |
| Pontifex et al., [148] | 34 | Within | Running | Moderate | 20 | Attention | Pre-Post | 2 | 0 | −0.39, 0.4 |
| Schmidt-Kassow et al., [149] | 18; 31 | Within | Walking | Light | 30 | Memory | During | 2 | 0.05 | −0.36, 0.43 |
| Tsai et al., [150] | 60 | Between | Resistance | Moderate Vigorous | 45 | Executive function | Pre-Post | 4 | 0.02 | −0.33, 0.36 |
| Bantoft et al., [151] | 45 | Within | Walking | Light | <60 | Executive function / Attention / Information processing | During | 4 | −0.1 | −0.39, 0.17 |
| Larson et al., [152] | 69 | Between | Walking | Light | 60 | Executive function | During | 2 | −0.03 | −0.43, 0.35 |
| Osgood, [153] | 86 | Between | Sport Activity | Light | 2 | Attention | Post | 1 | −0.06 | −0.47, 0.37 |
| Perciavalle et al., [154] | 30 | Within | Cycling | Vigorous | Volitional Exhaustion | Executive function | Pre-Post | 4 | 0.22 | −0.11, 0.61 |
| Shia et al., [155] | 17 | Within | Cycling | Vigorous | 45 | Attention | Pre-Post | 1 | −0.11 | −0.55, 0.3 |
| Stevens et al., [156] | 35; 34 | Between | Cycling | Light Moderate | 15 | Learning | During | 4 | 0.04 | −0.31, 0.4 |
| Weng et al., [157] | 26 | Within | Cycling | Moderate | 30 | Executive function | Pre-Post | 4 | 0.29 | −0.07, 0.67 |
| Alloway et al., [158] | 72 | Within | Running | Light | 8 | Executive function | Pre-Post | 1 | −0.01 | −0.43, 0.42 |
| Ando et al., [159] | 14 | Within | Cycling | Moderate Vigorous | 6 | Perception | During | 2 | −0.08 | −0.48, 0.31 |
| Brush et al., [160] | 28 | Within | Resistance | Light Vigorous | 45 | Executive function | Post | 60 | −0.1 | −0.26, 0.04 |
| Connell et al., [161] | 24 | Between | Cycling | Moderate | 180 | Attention | Pre-Post | 4 | −0.21 | −0.6, 0.12 |
| Hsieh et al., [162] | 20 | Within | Resistance | Moderate Vigorous | 30 | Executive function | Pre-Post | 1 | 0.04 | −0.39, 0.46 |
| Hsieh et al., [163] | 18 | Within | Resistance | Moderate | 30 | Executive function | Post | 1 | 0.03 | −0.39, 0.47 |
| Komiyama et al., [16] | 10 | Within | Cycling | Moderate | 30 | Executive function | During | 3 | 0.04 | −0.33, 0.42 |
| Lowe et al., [164] | 51 | Within | Walking | Light Moderate | 20 | Executive function | Pre-Post | 4 | 0.01 | −0.34, 0.35 |
| Torbeyns et al., [165] | 23 | Within | Cycling | Light | 30 | Memory / Executive function / Attention | During | 6 | −0.1 | −0.42, 0.22 |

**Table 1 (continued) | Study description and estimated effect sizes**

| Study | N | Experimental design | Exercise type | Exercise intensity | Duration (min) | Cognitive domain | Test time | N effects | g | 89% HDI |
|---|---|---|---|---|---|---|---|---|---|---|
| Tsukamoto et al.,[166] | 12 | Within | Cycling | Moderate Vigorous | 40 | Executive function | Pre-Post | 16 | −0.19 | −0.45, 0.07 |
| Zach & Shalom,[167] | 20 | Within | Running Volleyball Resistance | Vigorous | 15; Game duration | Executive function | Pre-Post | 6 | 0.37 | 0.02, 0.7 |
| Chang et al.,[168] | 30 | Within | Cycling | Vigorous | 30 | Executive function | Post | 1 | 0.35 | −0.07, 0.9 |
| Crush & Loprinzi,[169] | 352 | Between | Running | Moderate | 10; 20; 30; 45; 60 | Executive function Attention | Post | 75 | −0.14 | −0.28, 0.0 |
| González Fernández et al.,[170] | 18 | Within | Cycling | Light | 45 | Attention | During | 1 | 0.02 | −0.39, 0.45 |
| Lindheimer et al.,[171] | 60 | Between | Cycling | Light | 25 | Executive function | Pre-Post | 16 | −0.12 | −0.35, 0.12 |
| Lowe et al.,[172] | 28 | Within | Walking | Light Moderate | 20 | Executive function | Pre-Post | 4 | −0.31 | −0.68, 0.06 |
| Luu & Hall,[173] | 31 | Within | Yoga | Light | 25 | Executive function | Pre-Post | 2 | −0.19 | −0.62, 0.19 |
| Randolph & O'Connor,[174] | 18 | Within | Walking | Light | 10 | Attention Executive function Information processing | Pre-Post | 28 | −0.06 | −0.25, 0.14 |
| Slutsky et al.,[175] | 24 | Between | Cycling | Light | 15 | Attention Executive function | Pre-Post | 7 | −0.18 | −0.47, 0.14 |
| Sudo et al.,[176] | 32 | Between | Cycling | Vigorous | Volitional exhaustion | Executive function | Pre-Post | 3 | 0.28 | −0.07, 0.7 |
| Cuttler et al.,[177] | 120 | Between | Resistance walking | Light moderate | 30 | Memory Attention | Pre-Post | 6 | −0.61 | −0.98, −0.25 |
| Daikoku et al.,[178] | 44 | Between | Cycling | Light | 3 | Memory | During | 1 | −0.04 | −0.5, 0.35 |
| Elkana et al.,[179] | 69 | Between | Cycling | Moderate | 15 | Executive function | Pre-Post | 1 | −0.03 | −0.5, 0.35 |
| Fenesi et al.,[180] | 77 | Between | Calisthenics | Light | 13 | Learning Memory | Post | 2 | 0.16 | −0.24, 0.55 |
| Kendall,[181] | 60 | Between | HIIT | Vigorous | 20 | Information processing Learning Cognitive Control | Post | 8 | 0.82 | 0.42, 1.16 |
| Legrand et al.,[182] | 101 | Between | Running | Moderate | 15 | Attention Executive function | Pre-Post | 2 | 0.05 | −0.34, 0.44 |
| Samani & Heath,[183] | 14 | Within | Cycling | Vigorous | 10 | Executive function | Pre-Post | 1 | 0.08 | −0.34, 0.52 |
| Siddiqui & Loprinzi,[184] | 20 | Within | Walking | Moderate | 20 | Memory Executive function | Pre, During, Post | 4 | 0.02 | −0.31, 0.4 |
| Sng et al,[185] | 80 | Between | Walking | Light | 15 | Memory | During, Post | 4 | −0.25 | −0.62, 0.1 |
| Wade & Loprinzi,[186] | 34 | Between | Walking | Moderate | 15 | Memory | Post | 1 | −0.13 | −0.56, 0.3 |
| Yamazaki et al.,[187] | 30 | Within | Cycling | Light Moderate | 10 | Attention Executive function | Pre-Post | 16 | 0.07 | −0.16, 0.32 |
| Baker et al.,[188] | 19 | Within | Cycling | Light | 120 | Attention Executive function | During | 3 | −0.06 | −0.44, 0.31 |
| Du Rietz et al.,[189] | 26 | Within | Cycling | Vigorous | 20 | Attention Executive function | Pre-Post | 4 | −0.06 | −0.42, 0.28 |
| Engeroff et al.,[190] | 26 | Within | Resistance | Moderate Vigorous Maximal | 60 | Executive function | Pre-Post | 4 | 0.1 | −0.26, 0.45 |

**Table 1 (continued) | Study description and estimated effect sizes**

| Study | N | Experimental design | Exercise type | Exercise intensity | Duration (min) | Cognitive domain | Test time | N effects | g | 89% HDI |
|---|---|---|---|---|---|---|---|---|---|---|
| Haynes et al., [191] | 24 | Within | Walking | Light | 15 | Memory | During, Post | 9 | −0.04 | −0.32, 0.24 |
| Johnson & Loprinzi, [192] | 40 | Within | Walking | Moderate Vigorous | 15 | Memory | Post | 2 | −0.01 | −0.04, 0.38 |
| McGowan et al., [193] | 58 | Within | Running | Moderate | 20 | Executive function | Pre-Post | 2 | −0.02 | −0.39, 0.39 |
| Mehren et al., [107] | 31; 32 | Within | Cycling | Vigorous | 30 | Executive function | Post | 8 | 0.15 | −0.14, 0.45 |
| Schmidt et al., [194] | 15; 19 | Within | Running | Vigorous | 38 | Attention | Pre-Post | 6 | −0.24 | −0.57, 0.09 |
| Stenling et al., [195] | 32 | Within | Walking | Moderate | 3 | Executive function Attention | Post | 8 | 0 | −0.29, 0.29 |
| Wu et al., [196] | 35 | Within | Cycling Resistance | Light Vigorous | 30 | Executive function | Post | 8 | 0.28 | −0.03, 0.59 |
| Zhou & Qin, [197] | 72 | Between | Cycling | Light | 25 | Executive function | Pre-Post | 2 | −0.05 | −0.42, 0.37 |
| Aly & Kojima, [198] | 40 | Between | Cycling | Light | 20 | Executive function | Pre-Post | 2 | 0.07 | −0.32, 0.47 |
| Chacko et al., [199] | 15 | Within | Cycling | Vigorous | 30 | Executive function | Pre-Post | 2 | 0.06 | −0.32, 0.49 |
| Kao et al., [8] | 23 | Within | Walking | Light | 20 | Executive function | Post | 4 | 0.24 | −0.10, 0.63 |
| Morris et al., [200] | 14 | Within | Cycling | Moderate | 30 | Attention Executive function | Pre-Post | 6 | −0.01 | −0.33, 0.3 |
| Walsh et al., [201] | 15; 13 | Within | Walking | Moderate | 20 | Executive function Attention | Pre-Post | 8 | 0.01 | −0.31, 0.29 |
| Kao et al., [108] | 36 | Within | HIIT Walking | Moderate | 16 | Executive function | Post | 2 | 0.32 | −0.06, 0.78 |
| Kim et al., [202] | 16 | Within | Walking | Vigorous | 10 | Executive function | Pre-Post | 2 | −0.03 | −0.38, 0.41 |
| Klatt & Smeeton, [203] | 27 | Within | Cycling | Vigorous | Task Completion | Memory | During | 4 | −0.15 | −0.5, 0.21 |
| Kuhne et al., [204] | 50 | Between | Cycling | Moderate | 40-55 | Memory | Post | 2 | −0.09 | −0.49, 0.29 |
| Manocchio & Lowe, [205] | 22 | Within | Walking | Light Moderate | 20 | Attention Executive function | Pre-Post | 8 | −0.06 | −0.34, 0.25 |
| Miyashiro et al., [206] | 17 | Within | Push-Ups | Vigorous | 20 | Executive function | Post | 1 | 0 | −0.45, 0.41 |
| Trammell & Aguilar, [207] | 28 | Within | Running | Vigorous | 20 | Memory Executive function | Pre-Post | 5 | −0.11 | −0.47, 0.2 |
| Zhu et al., [208] | 16 | Within | Cycling Running | Moderate Vigorous | 40 | Executive function | Pre-Post | 16 | 0.1 | −0.14, 0.34 |
| Aguirre-Loaiza et al., [209] | 19 | Between | Cycling | Vigorous | 20 | Executive function | Pre-Post | 4 | −0.06 | −0.41, 0.3 |
| Drollette & Meadows, [210] | 22 | Within | HIIT | Vigorous | 9 | Executive function | Post | 6 | 0.06 | −0.24, 0.39 |
| Engeroff et al., [211] | 26 | Within | Resistance | Moderate Vigorous | 60 | Attention | Pre-Post | 16 | −0.17 | −0.39, 0.08 |
| Frith et al., [212] | 45 | Within | Running | Moderate | 15 | Executive function | During | 1 | 0.06 | −0.4, 0.46 |
| Kao et al., [213] | 27 | Within | Running | Moderate | 24 | Attention Executive function | Pre-Post | 5 | −0.04 | −0.35, 0.31 |
| LaCount et al., [214] | 18 | Within | HIIT | Vigorous | 16 | Attention Executive function Information processing | Post | 3 | −0.09 | −0.46, 0.28 |
| Loprinzi & Storm, [215] | 180; 225; 158 | Between | Treadmill | Moderate Vigorous | 25 | Memory | Post | 6 | −0.02 | −0.33, 0.3 |

**Table 1 (continued) | Study description and estimated effect sizes**

| Study | N | Experimental design | Exercise type | Exercise intensity | Duration (min) | Cognitive domain | Test time | N effects | g | 89% HDI |
|---|---|---|---|---|---|---|---|---|---|---|
| Shirzad et al., [216] | 28 | Within | Cycling | Light | 26 | Executive function | Pre-Post | 2 | −0.23 | −0.64, 0.17 |
| Zheng et al., [217] | 27 | Within | Cycling | Moderate | 15 | Executive function | During | 2 | 0.27 | −0.11, 0.72 |
| Chueh et al., [218] | 30 | Within | Step Exercise | Vigorous | 20 | Attention Executive function | Pre-Post | 8 | 0.15 | −0.15, 0.44 |
| Yamada et al., [219] | 85 | Within | Recumbent Bike | Vigorous | 20 | Executive function | Pre-Post | 4 | 0.02 | −0.31, 0.38 |
| B. Zhang et al., [220] | 18 | Within | Cycling | Moderate | 25 | Perception | Post | 4 | 0.29 | −0.09, 0.65 |
| D. Zhang et al., [221] | 76 | Within | Cycling | Light | 20 | Attention | Pre-Post | 4 | 0.03 | −0.33, 0.36 |

(BF = 2.78), and standard normal (BF = 5.27) priors. Estimates of between-study heterogeneity were also robust across the NE prior ($\tau_{between} = 0.29 \pm 0.05$; HDI = [0.20, 0.37]), the PE prior ($\tau_{between} = 0.29 \pm 0.05$; HDI = [0.21, 0.37]), and $t$ likelihood function ($\tau_{between} = 0.31 \pm 0.03$; HDI = [0.26, 0.38]) (Fig. 5b). In contrast, within study heterogeneity was estimated to be lower when using the $t$ likelihood function ($\tau_{within} = 0.17 \pm 0.02$; HDI = [0.13, 0.19]) relative to the NE ($\tau_{within} = 0.65 \pm 0.03$; HDI = [0.61, 0.70]) and PE ($\tau_{within} = 0.65 \pm 0.02$; HDI = [0.60, 0.70]) priors (Fig. 5c). Note, this reduction reflects the diminished influence of outliers on variance estimates by the inclusion of the shape parameter for the $t$ distribution ($v = 1.52 \pm 0.14$; HDI = [1.30; 1.73]). In addition to testing the robustness of parameter estimates, a model comparison was conducted to determine if either the null or positive effect prior was more probable given the data. The $t$-likelihood function was not included in this comparison since it would only indicate if effect sizes were more likely to have been drawn from either a normal or $t$-distribution. When compared to a standard normal prior, there was anecdotal evidence in favor of both the PE (BF = 2.56) and NE (BF = 1.48) priors. Relative to the PE prior, there was anecdotal evidence against the NE prior (BF = 0.73). Altogether, parameter estimates were not biased by the prior or likelihood function.

## Executive function meta-analysis

Considering that the majority of the effect sizes were from tasks that probed executive function, and that this cognitive domain encompasses multiple sub-domains, a separate meta-analysis and set of meta-regressions were conducted on this subset of data. Categorization criteria from previous meta-analyses and systematic reviews[17,24,26] were used to classify effect sizes into the following sub-domains of executive function: working memory, cognitive control, decision making, planning, and inhibition. For completeness, the primary moderators used in the main meta-analysis were also tested.

The results were similar to the main meta-analysis. There was very strong evidence in favor of exercise having a small positive influence on overall task performance ($g = 0.20 \pm 0.06$; HDI = [0.12, 0.30]; BF = 29.57), and a moderate degree of heterogeneity both within ($\tau_{within} = 0.51 \pm 0.03$; HDI = [0.47, 0.57]) and between studies ($\tau_{between} = 0.40 \pm 0.06$; HDI = [0.30, 0.48]). Subgroup analyses indicated that a model including the moderator task outcome measure had more predictive power relative to a null counterpart ($BF_{Inclusion} = 48.43$). Paralleling the main meta-analysis, there was very strong evidence that acute exercise improved RT on executive function tasks ($g = 0.32$; HDI = [0.21, 0.42]; BF = 748.18), but no credible evidence was observed for an effect on accuracy ($g = 0.13$; HDI = [0.04, 0.23]; BF = 0.63) (Table 5). Furthermore, there was moderate evidence in favor of a positive impact of exercise on inhibition ($g = 0.21$; HDI = [0.09, 0.33]; BF = 3.14) and working memory ($g = 0.22$; HDI = [0.11, 0.34]; BF = 6.89) (Fig. 6). Yet, a model including executive function sub-domain as a moderator did not improve model performance ($BF_{Inclusion} = 7.52 \times 10^{-4}$), nor did models including interactions between moderators.

## Discussion

A large corpus of empirical work has examined how a single bout of acute exercise modulates activity within multiple brain systems that underly cognition. Despite inconsistencies in results across empirical studies, there is consensus amongst previous reviews and meta-analyses that acute exercise impacts behavioral performance[18,20,22] and that this relationship is moderated by both exercise protocol and behavioral task characteristics. The goal of the present work was to address two key limitations of previous meta-analyses. First, recent meta-analyses have a narrower focus, often limited to a single cognitive domain or a specific subset of domains. In contrast, the current meta-analysis presents an updated synthesis of the literature spanning a much wider range of cognitive domains. Second, in contrast to previous frequentist approaches, a Bayesian framework was adopted allowing for the quantification of the degree of evidence in favor of the hypothesis that acute exercise influences cognition in young healthy adults.

## Table 2 | Primary moderator estimates

| Exercise moderator variable | N | g | 89% HDI | BF$_{10}$ | Cognitive moderator variable | N | g | 89% HDI | BF$_{10}$ |
|---|---|---|---|---|---|---|---|---|---|
| Intensity | | | | | Domain | | | | |
| Light | 167 | 0.10 | −0.02, 0.22 | 0.13 | Attention | 109 | 0.06 | −0.08, 0.17 | 0.08 |
| Moderate | 222 | 0.07 | −0.03, 0.18 | 0.09 | Executive function | 434 | 0.18 | 0.10, 0.27 | 36.97 |
| Vigorous | 253 | 0.19 | 0.09, 0.28 | 5.03 | Information processing | 15 | 0.12 | −0.17, 0.41 | 0.14 |
| | | | | | Learning | 12 | 0.24 | −0.11, 0.59 | 0.25 |
| Duration (minutes) | | | | | Memory | 44 | −0.06 | −0.25, 0.13 | 0.08 |
| ≤16 | 161 | 0.14 | 0.02, 0.26 | 0.30 | Motor skills | 6 | −0.03 | −0.51, 0.49 | 0.20 |
| 20–27 | 152 | 0.15 | 0.02, 0.26 | 0.32 | Perception | 22 | 0.13 | −0.17, 0.44 | 0.15 |
| 30–35 | 93 | 0.08 | −0.08, 0.22 | 0.09 | | | | | |
| 40–45 | 113 | 0.04 | −0.13, 0.21 | 0.09 | | | | | |
| >60 | 48 | −0.03 | −0.21, 0.15 | 0.08 | Task outcome | | | | |
| Not provided | 75 | 0.37 | 0.17, 0.57 | 6.21 | Accuracy | 377 | 0.04 | −0.04, 0.12 | 6.15e-2 |
| | | | | | Reaction time | 265 | 0.27 | 0.18, 0.36 | 6.71e3 |
| Type circuit | 8 | 0.08 | −0.33, 0.53 | 0.26 | | | | | |
| Cycling | 204 | 0.21 | 0.11, 0.32 | 14.74 | Task completion time (relative to exercise) | | | | |
| HIIT | 18 | 0.73 | 0.40, 1.09 | 26.05 | During | 82 | 0.02 | −0.18, 0.13 | 0.09 |
| Resistance | 97 | −0.06 | −0.29, 0.14 | 0.11 | Immediately after | 315 | 0.16 | 0.11, 0.30 | 4.03 |
| Running | 172 | 0.05 | −0.10, 0.19 | 0.08 | 20–75 min post | 94 | 0.22 | 0.13, 0.44 | 0.76 |
| Sport activity | 26 | 0.04 | −0.23, 0.29 | 0.11 | >180 min post | 151 | 0.08 | −0.03, 0.28 | 0.10 |
| Walking | 117 | 0.04 | −0.10, 0.19 | 0.07 | | | | | |

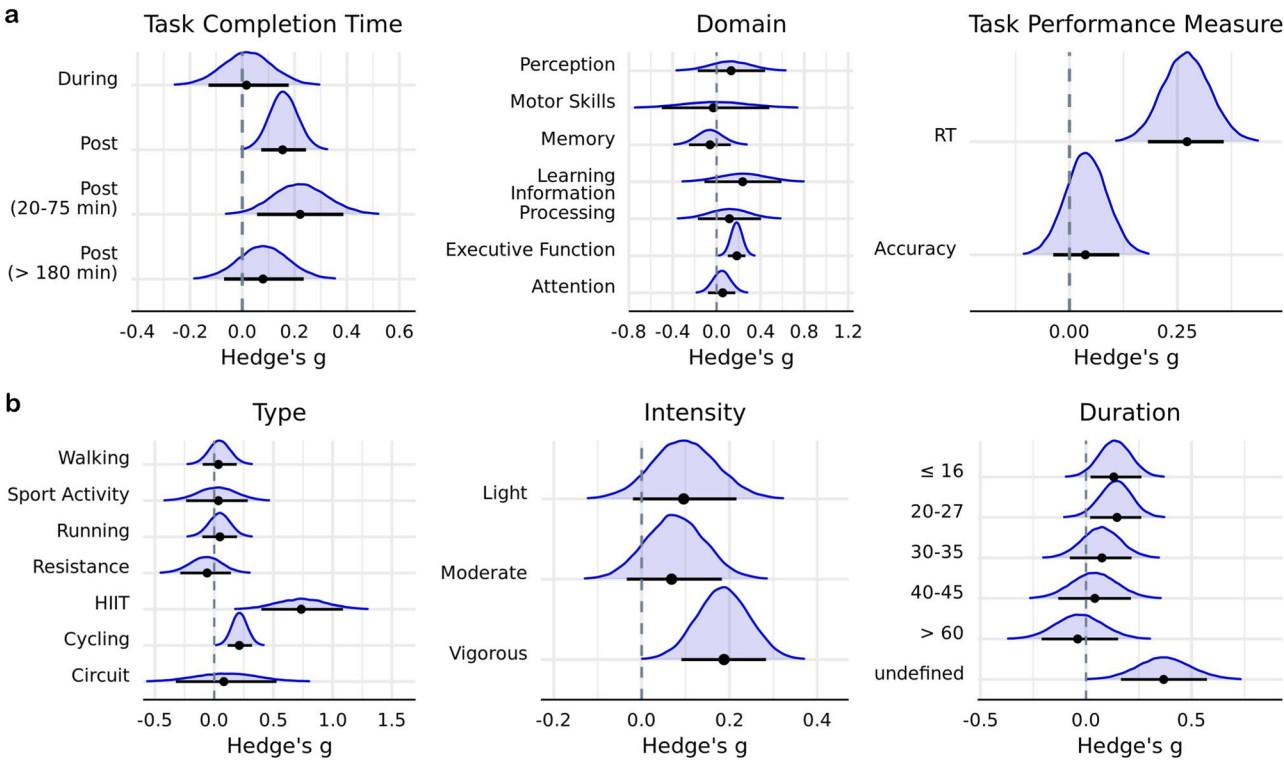

**Fig. 3 | Subgroup analyses.** Posterior distributions of **a** cognitive and **b** exercise moderators. Horizontal black line indicates the 89% HDI interval, while the black dot represents the mode of the posterior distribution. Intervals derived using n = 651 effect sizes.

The current meta-analysis observed that acute exercise has a small positive influence on overall cognitive task performance, and sensitivity analyses indicated that the alternative hypothesis was 6.51–8.77 times more likely than the null across multiple priors and likelihood functions. The magnitude and directionality of this effect were consistent with the results of previous meta-analyses on acute exercise and cognition[18,21,22,62]. Subgroup analyses suggested that this relationship is moderated by task performance measure, cognitive domain, exercise type and intensity, and the time of task completion relative to exercise cessation. Model comparison results indicated that accounting for variations amongst moderator levels did not improve

## Table 3 | Secondary moderator estimates

| Moderator variable | N | g | 89% HDI | BF$_{10}$ |
|---|---|---|---|---|
| Publication year | 642 | 0.13 | 0.06, 0.20 | 3.25 |
| Exp. design | | | | |
| Between | 193 | 0.03 | −0.11, 0.16 | 0.07 |
| Within | 449 | 0.17 | 0.08, 0.24 | 12.18 |
| Age (years) ($\mu = 22.49$) | 599 | 0.14 | 0.06, 0.21 | 3.13 |
| % Female | 577 | 0.12 | 0.05, 0.20 | 4.23 |
| BMI (kg/m$^2$) ($\mu = 24.02$) | 378 | 0.20 | 0.08, 0.30 | 1.28 |
| VO2 max (ml/kg/min) ($\mu = 43.22$) | 299 | 0.19 | 0.05, 0.34 | 1.08 |
| Height (cm) ($\mu = 158.55$) | 275 | 0.21 | 0.09, 0.33 | 2.90 |
| Weight (kg) ($\mu = 65.02$) | 283 | 0.21 | 0.08, 0.33 | 4.28 |

## Table 4 | Subgroup model comparisons

| Model | BF$_{Inclusion}$ |
|---|---|
| No moderators | 1 |
| Exercise intensity | 2.16e-4 |
| Exercise duration | 9.35e-4 |
| Exercise type | 2.00e-2 |
| Cognitive domain | 4.97e-3 |
| Task outcome | 357.10 |
| Task completion time | 5.52e-4 |
| Publication date | 9.05e-4 |
| Experimental design | 0.04 |
| Age | 0 |
| % Female | 4.53e-3 |
| BMI | 0.04 |
| VO2 | 8.40e-3 |
| Height | 2.89e-3 |
| Weight | 0.01 |

predictive performance. Given our eligibility criteria, these results are limited to healthy individuals between the ages of 18–45 years old.

Similar to McMorris et al.[63], acute exercise was found to improve RT but no credible evidence was observed for an influence on accuracy. A possible explanation for this differential impact on task outcome measures is that exercise modulates primary motor cortex (M1) excitability[64]. There is accumulating evidence that acute exercise increases M1 intracortical facilitation[65–68] and inhibition[69,70]. Yamazaki et al.[68] observed that the intracortical circuits of both exercised (i.e., legs) and non-exercised (i.e., hand) effectors are modulated by an acute bout of low intensity pedaling. Thus, alterations in the activity of excitatory or inhibitory circuits of non-exercised cortical representations may promote faster RT on cognitive tasks. However, the lack of concurrent changes in corticospinal excitability or motor-evoked potentials suggests that this explanation is not a viable account of a mechanism that engenders faster RTs. An alternative explanation is that exercise increases peripheral and central concentrations of catecholamines, such as norepinephrine, epinephrine, and dopamine, which in turn improves the speed of cognition[1,71,72]. Indeed, acute exercise has been found to improve response time on choice RT, decision-making, and interference tasks[18,73,74]. Yet, it is unclear as to why changes in neurochemical levels would facilitate RT but have no impact on accuracy. Considering that physical activity modulates population-level tuning in the sensory areas of nonhuman animals and invertebrates[75–81], along with sensory responses in humans[82–84], it stands to reason that the fidelity of stimulus representations would also be impacted, resulting in changes in

accuracy. Changes in the fidelity of feature selective stimulus representations can be determined by applying encoding models to recorded neural activity[83,85–90]. For instance, Garrett et al.[91] applied an inverted encoding model to topographical patterns of alpha band activity, recorded at the scalp, while subjects completed a spatial working memory task both at rest and during a bout of moderate intensity cycling. Notably, it was possible to reconstruct spatially selective responses during exercise, and the selectivity of these responses decreased during exercise relative to rest. Therefore, encoding models can be a powerful tool for future research to demystify how the precision of task-relevant representations is influenced by exercise. It is also important to keep in mind that many psychological tasks are relatively simple to do, which can lead to ceiling effects that may mask the influence of exercise on accuracy measures. Lastly, the differential impact of exercise on accuracy and RT may be due to the relative sensitivities of these dependent measures to modulations of different stages of information processing. For example, there is evidence that in near-threshold tasks accuracy is sensitive to perceptual manipulations, whereas in supra-threshold (i.e., perceptually easy tasks, including many of those used in the studies in this meta-analysis) RT is sensitive to modulations in both perceptual and post-perceptual processes[92,93]. Indeed, Davranche et al.[73] utilized a drift diffusion model to determine which aspects of decision-making are modulated by HIIT. Importantly, drift rate and decision response boundary size increased significantly after exercise relative to before, while non-decision time decreased. This suggests there was an improvement in perceptual discrimination, the efficiency of non-decisional processes (e.g., motor execution), and the adoption of a more conservative criterion. Future research employing computational models of response time and representational fidelity is needed to develop a comprehensive understanding of the selective influence exercise on information processing speed and accuracy.

Parameter estimates of a model including exercise modality as a moderator suggested that engaging in cycling or HIIT may beneficially impact cognition, especially on attentional and executive processes. Cycling is a commonly used modality in exercise and cognition research. Numerous empirical studies have found that a bout of cycling benefits inhibition, as measured using either the Stroop or Eriksen Flanker task[15,94–99]. Improvements in planning[94,100], task-switching[27,101,102], and the speed of decision making[103] have also been reported. In contrast to the ubiquity of cycling, the use of HIIT workouts in exercise and cognition research is a relatively recent practice, hence the small number of effect sizes from studies using this modality compared to other types of exercise. The number of effect sizes is important because low-level parameters in a hierarchical model are influenced both by the subset of data directly dependent on the low-level parameter, and by high-level parameter estimates that rely on all of the data. This makes low-level parameter estimates indirectly dependent on the entire dataset, and causes shrinkage in estimates at all levels of the model. In other words, the estimated relationship between HIIT and behavioral performance is derived directly from the few representative effect sizes and indirectly from the rest of the data. The observed positive effect of HIIT on cognition corroborates previous findings. For example, Alves et al.[3] observed that the time to complete a Stroop Task decreased after ten 1-minute bouts of exercising at 80% heart rate reserve relative to a control condition. Improvements in time-dependent measures on interference tasks (i.e., Stroop and flanker) have been correlated with an increase in left dorsolateral prefrontal cortex activity, as measured with functional near infrared spectroscopy (fNIRS), and a decrease in P3 latency measured with EEG[104]. Furthermore, enhancements have also been shown to coincide with an increase in peripheral levels of neural growth factors and lactate[105]. Lastly, a recent meta-analysis on elite athletes observed that HIIT team-based sports had a positive impact on cognitive task performance[25]. Interestingly, because of the small number of published studies in the literature, it is currently unclear if the type of exercise modality used for HIIT workouts (e.g., cycling, sprinting, resistance) differentially impacts cognition.

Behavioral task performance was found to be improved by engaging in vigorous intensity exercise. These results are surprising, considering that exercise intensity is believed to have an inverted-U relationship with

**Fig. 4 | Interactions between cognitive and exercise moderators.** Posterior mode estimates of models including interactions between cognitive domain and **a** exercise type and **b** task outcome measure. Width of line represents 89% HDI derived using n = 651 effect sizes.

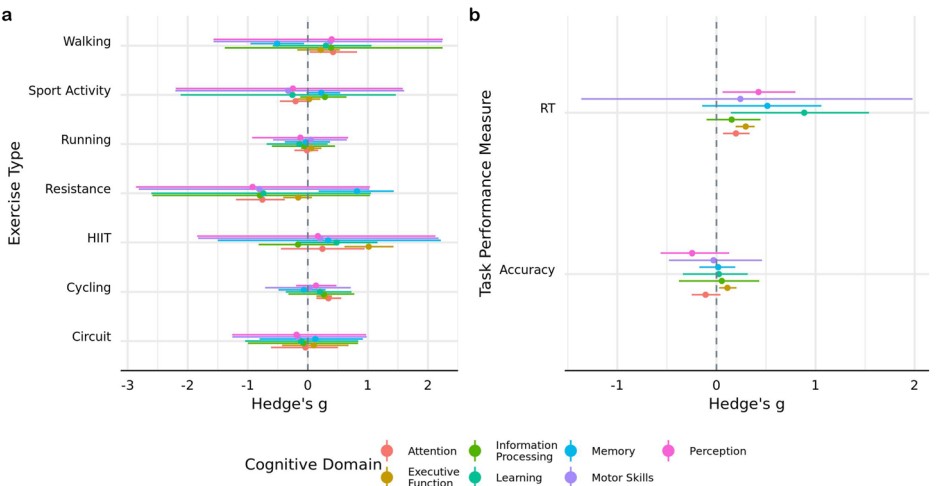

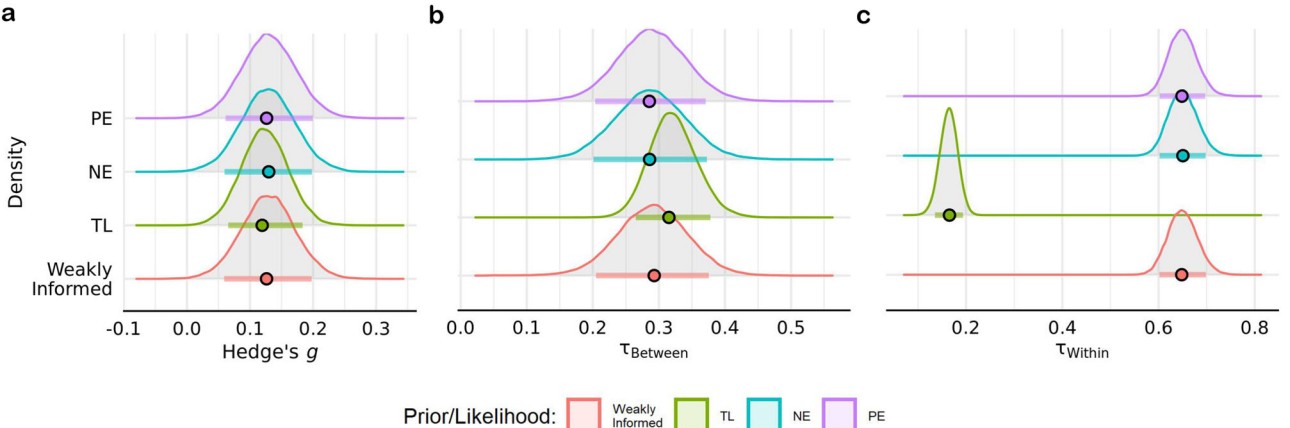

**Fig. 5 | Sensitivity analyses.** Estimates for the **a** overall pooled effect size, **b** between- and **c** within-study heterogeneity parameters across the t-likelihood function (TL), weakly informed, null effect (NE), and positive effect (PE) priors. Color dots represent mode of posterior distributions, while color horizontal line depicts the 89% HDI derived using $n = 651$ effect sizes.

performance; where moderate intensity exercise elicits the greatest enhancements while more intense, fatiguing exercise imposes decrements[18,62,63,71,106,107]. This effect could be driven by HIIT workouts, but may also depend on multiple cognitive task and exercise protocol characteristics. For instance, Chang et al.[18] observed that exercise intensity was only a significant moderator when cognition was tested post-exercise. Similarly, Oberste et al.[23] found that exercise intensity influenced time-dependent measures of interference control but not accuracy. When considering these results, one must also consider that both aforementioned meta-analyses included studies whose subjects were children, adolescents, and older adults. In contrast, the current study was limited to young adults, and there is evidence that the effect of exercise on cognition is comparatively smaller in this age group[18,23]. Thus, a model containing an interaction between cognitive domain, task outcome measure, and age groups across the lifespan may be required to observe evidence for an effect of intensity. In addition, there was evidence for the enhancing effects of exercise post-cessation, corroborating previous research[1,18,94]. Interestingly, in the current meta-analysis cognition was not found to be impacted during exercise. Prior meta-analytic findings on cognition during exercise are mixed, with some reporting that it is exacerbated[20], while others that find evidence for an enhancement[18].

Given that the majority of the effect sizes were from tasks that probed executive function, a separate meta-analysis was conducted on this subset of data. This analysis revealed that exercise has a small positive impact on RT

measures of executive processes. When looking at model parameters, there was evidence in favor of exercise-enhancing inhibition and working memory. Behavioral research has shown that both the accuracy[9] and speed of working memory[108,109] are facilitated by an instance of physical activity. What remains to be determined is the neural mechanisms that engender these behavioral effects. Kao et al.[108] observed that a reduction in RT on the Sternberg task post-HIIT corresponded to an increase in frontal alpha desynchronization during encoding, maintenance, and retrieval periods when working memory load is high. Neuroimaging studies have also found evidence for changes in the activation levels of frontal areas[110] and their connectivity with the intraparietal sulcus post-exercise[111]. These changes in neural activity were not accompanied by a change in behavior, suggesting that more research is needed to demystify the neuromodulatory effect of acute exercise on working memory.

Engaging in repeated bouts of acute exercise over a long period of time can have lasting changes on baseline neurochemical levels, cortical volume, and structural/functional connectivity, which can alter cognitive task performance[1,112–115]. Research investigating the influence of these long-term interventions on cognition has primarily focused on children or older adults. Systematic reviews and meta-analyses suggest that exercise has a small to moderate beneficial impact on general task performance for both of these age groups, with the largest effect sizes observed for measures of executive function, attention, and academic performance[35]. Despite the relative paucity of meta-analyses on how exercise interventions impact cognition in

**Table 5 | Executive function moderator estimates**

| Exercise moderator variable | N | g | 89% HDI | BF$_{10}$ | Cognitive moderator variable | N | g | 89% HDI | BF$_{10}$ |
|---|---|---|---|---|---|---|---|---|---|
| Intensity | | | | | Sub-domain | | | | |
| Light | 107 | 0.23 | 0.08, 0.37 | 1.71 | Cognitive control | 55 | 0.23 | 0.06, 0.39 | 0.73 |
| Moderate | 151 | 0.15 | 0.01, 0.28 | 0.30 | Decision making | 28 | 0.10 | −0.16, 0.39 | 0.14 |
| Vigorous | 175 | 0.24 | 0.12, 0.36 | 13.49 | Inhibition | 153 | 0.21 | 0.09, 0.33 | 3.14 |
| | | | | | Planning | 18 | 0.14 | −0.12, 0.38 | 0.15 |
| Duration (minutes) | | | | | Working memory | 179 | 0.22 | 0.11, 0.34 | 6.89 |
| ≤16 | 93 | 0.25 | 0.09, 0.40 | 1.76 | | | | | |
| 20–27 | 99 | 0.13 | −0.02, 0.27 | 0.19 | Task outcome | | | | |
| 30–35 | 68 | 0.19 | 0.03, 0.36 | 0.44 | Accuracy | 253 | 0.13 | 0.04, 0.23 | 0.63 |
| 40–45 | 100 | 0.13 | −0.06, 0.32 | 0.16 | Reaction time | 180 | 0.32 | 0.21, 0.42 | 749.18 |
| >60 | 25 | 0.11 | −0.12, 0.32 | 0.13 | | | | | |
| Not provided | 48 | 0.53 | 0.29, 0.83 | 24.40 | Type | | | | |
| | | | | | Circuit | 8 | 0.15 | −0.31, 0.65 | 0.31 |
| Task completion time (relative to exercise) | | | | | Cycling | 133 | 0.28 | 0.15, 0.42 | 19.58 |
| During | 39 | 0.23 | 0.004, 0.44 | 0.45 | HIIT | 12 | 0.96 | 0.56, 1.38 | 71.23 |
| Immediately after | 197 | 0.21 | 0.10, 0.31 | 4.17 | Resistance | 90 | −0.07 | −0.33, 0.17 | 0.12 |
| 20–75 min post | 72 | 0.31 | 0.13, 0.47 | 4.11 | Running | 116 | 0.06 | −0.12, 0.25 | 0.10 |
| >180 min post | 125 | 0.13 | −0.03, 0.29 | 0.17 | Sport activity | 11 | 0.30 | −0.07, 0.68 | 0.35 |
| | | | | | Walking | 63 | 0.14 | −0.05, 0.34 | 0.18 |

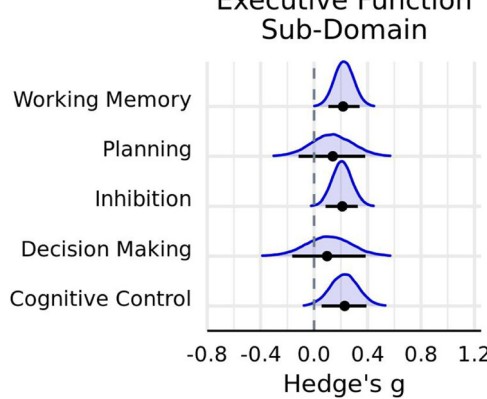

**Fig. 6 | Subgroup analyses of effects from tasks testing executive function.** Posterior distributions for executive function sub-domain. Horizontal black line indicates the 89% HDI interval, while the black dot represents the mode of the posterior distribution, which was derived using n = 433 effect sizes.

healthy young adults, recent work suggests that it may have a similar beneficial effect. Indeed, a recent meta-analysis, conducted by Ludyga et al.[116], indicated that long-term exercise interventions have a small positive influence on general cognition regardless of age. The magnitude of this effect was dependent on the interaction between intervention length and exercise duration, with longer interventions and sessions producing greater benefits. Integrating these findings with the current meta-analysis, there is support for the notion that the beneficial impact of long-term interventions on cognition may be a product of repeated exposure to acute exercise induced effects.

There are a number of possible explanations as to why exercise induced effects are small. One possibility is that cognitive function is at its peak during young adulthood, leaving little room for improvements in task performance. Indeed, previous reviews and meta-analyses have observed that the effect of exercise is moderated by age[35], with the greatest benefits observed for preadolescent children and older adults[18,23,26]. Contrary to this

account, though, the largest exercise induced effects were observed for executive processes, which are believed to be at peak efficiency during this period in the lifespan[117,118]. Furthermore, there was moderate evidence that the impact of exercise increased as the average age of sampled young adults also increased. Another explanation may be that cognition is resilient to slight or modest perturbations in overall global state. For example, Bullock et al.[119] demonstrated there was no change in accuracy or RT on a target detection task during experimentally induced hypoxia, hypercapnia, hypocapnia, and normoxia. Meta-analytic modeling of the influence of acute stress on executive function revealed that stress has a small negative impact on working memory and cognitive flexibility, but no impact on inhibition[120]. This suggests that cognition is able to selectively adapt to changes in physiological state caused by various types of stressors, including exercise. A final more intriguing and functional explanation for exercise having a small impact on cognition is that experimental protocols do not typically require the engagement of the body to execute the cognitive task, but rather have people engage in a cognitive task while exercising (or shortly thereafter). This experimental design contrasts real-world tasks that require engagement of the body in the service of the cognitive task. When components of the exercise are incorporated into task goals, then larger changes in performance may be observed. Empirical research investigating how exercise influences task performance in embodied settings versus classic laboratory settings (see[121] for review) is necessary to test the plausibility of this explanation. In addition, the notion that the integrated action of the body and the mind are required to produce the largest effects of exercise on cognition is consistent with a recent evolutionary account of the link between cognition and exercise[122].

The discrepancy in moderator results between the current meta-analysis and previous meta-analyses could be due to differences in the statistical approach. Frequentist methods typically conduct an omnibus test to determine if levels of a moderator are significantly different from one another and as a measure of a model's goodness of fit. In contrast, the Bayesian approach determines how likely the observed effect sizes are under a model that includes a moderator and if predictive power is increased. There are a few key advantages to using the Bayesian approach compared to classical frequentist methods. First, it models the uncertainty involved in estimates of between- and within-study heterogeneity and returns a full

posterior distribution for both parameters[123]. With these posterior distributions, one can simulate possible pooled effect sizes across credible levels of heterogeneity and develop an informed hypothesis for a subsequent meta-analysis. Similarly, the posterior distributions of effect size estimates can be used as well-informed prior distributions for new data. Importantly, this facilitates the updating of meta-analyses as new research is published. It should be mentioned that the degree of between-study heterogeneity was numerically similar to previous meta-analyses[18,22], implying that they did not suffer from an issue of underestimation by assuming heterogeneity to be a fixed quantity. Second, the Bayesian approach permits the inclusion of prior knowledge. Across all tested priors, there was evidence in favor of a pooled effect derived from averaging the reported estimates of previous meta-analyses. When comparing a prior distribution based on this knowledge to a null effect prior, the former was found to be more probable. Lastly, the posterior distribution of parameter estimates can be used to ascertain the likelihood that one will observe an effect size of a given magnitude for an exercise protocol and cognitive task combination. For example, a researcher could compute the probability that the influence of a bout of cycling on cognitive control will fall within the range of large effect sizes, even if that range does not encompass the maximum a posteriori probability estimate. In contrast, the frequentist approach only produces the maximum likelihood estimate and an interval around it based on fictitious repeats of the meta-analysis. Therefore, the Bayesian approach provides more information for designing future exercise and cognition studies.

## Limitations

A potential limitation in the current meta-analysis is the categorization of exercise type using the activity reported in each study. An alternative approach is to categorize exercise based on the theoretical and physiological distinctions between aerobic and anaerobic exercise. We did not adopt this approach here because many activities used in the literature typically include aerobic and anaerobic components, and basing their classification on what authors reported provides insights into the exercise modalities that have been predominantly used in the literature. Another limitation is the schema used to categorize exercise duration. In the event that a study did not report how long participants engaged in exercise, these effects were classified as "not provided", rendering them as uninterpretable. Lastly, sensitivity analyses were not conducted for moderator parameter estimates due to the high degree of computational demands. However, considering that the pooled effect size estimate was robust across multiple priors and likelihood functions, it is likely that moderator parameter estimates are also consistent.

## Conclusions

In summary, the current meta-analytic examination has shown that there is moderate evidence for an acute bout of aerobic exercise inducing a small enhancement in overall performance on cognitive tasks, especially on those that probe executive function and measure response time. Incorporating computational models of decision-making processes, such as drift-diffusion or signal detection models, into exercise research may provide useful insights into the nature of speeded executive processes. Furthermore, testing performance in a real-world setting where individuals typically engage in physical activity may amplify exercise-induced effects.

## Data availability

Data have been made publicly available at https://github.com/jggarrett23/PACMAn.

## Code availability

Code has been made publicly available at https://github.com/jggarrett23/PACMAn.

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

## Acknowledgements

This work was funded by the U.S. Army Combat Capabilities Development Command Soldier Center Measuring and Advancing Soldier Tactical Readiness and Effectiveness (MASTR-E) program through award W911NF-19F-0018 under US Army Research Office contract W911NF-19-D-0001 for the Institute for Collaborative Biotechnologies. The views expressed in this article are solely those of the authors and do not reflect the official policies or positions of the Department of Army, the Department of Defense, or any other department or agency of the U.S. government. The funders had no role in study design, data collection, and analysis, the decision to publish, or preparation of the manuscript. Thank you to Grace Giles, Ph.D., Julie Cantelon, Ph.D., and Neil Dundon, Ph.D., for providing guidance in conducting analyses and interpreting results. Lastly, thank you to Emily Machniak and Riddhima Chandra for assisting in data collection efforts.

## Author contributions

J.G., C.C., T.B., and B.G., designed the study, created the literature search criterion, and performed initial article screenings. J.G., C.C., and B.G. performed screenings of articles based on their full text. J.G., extracted effect sizes, conducted statistical analyses, and wrote the initial draft of the manuscript. B.G., T.B., and C.C. provided edits to the manuscript.

## Competing interests

The authors declare no competing interests.
