## [Peer Review File · Communications Psychology]

14th Sep 23

Dear Mr Garrett,

Thank you for your patience during the peer-review process, I apologize for the delay in processing the manuscript. Your manuscript titled "Acute Physical Activity has Selective Effects on Cognition" has now been seen by 2 reviewers, and I include their comments at the end of this message. They find your work of interest, but raised some important points. We are interested in the possibility of publishing your study in Communications Psychology, but would like to consider your responses to these concerns and assess a revised manuscript before we make a final decision on publication.

We therefore invite you to revise and resubmit your manuscript, along with a point-by-point response to the reviewers. Please highlight all changes in the manuscript text file.

Your revision must engage with all of the concerns voiced by the referees. Editorially, we consider especially the response to the following issues a precondition for further consideration of your work:

- 1) Both reviewers raised questions on the appropriateness of the moderator variable categorisations
- 2) Both reviewers raised questions on the eligibility criteria used in the analysis
- 3) Reviewer #2 raised the issue of the timeliness of the papers included (i.e., papers from 2020-2023 were not included).
- 4) Reviewer #1 raised the issue of publication bias. Although you report a funnel plot and Egger's regression intercept, we ask you to revisit this point. There are various frequentist and bayesian approaches available for estimating effect sizes that account for publication bias

Please use the following link to submit your revised manuscript, point-by-point response to the referees' comments (which should be in a separate document to any cover letter) and the completed checklist:

[link redacted]

Please do not hesitate to contact me if you have any questions or would like to discuss these revisions further. We look forward to seeing the revised manuscript and thank you for the opportunity to review your work.

Best regards,

Daniel Quintana

Daniel Quintana, PhD

Editorial Board Member

Communications Psychology

orcid.org/0000-0003-2876-0004

EDITORIAL POLICIES AND FORMATTING

Editorial Policy: Policy requirements (Download the link to your computer as a PDF.)

Furthermore, please align your manuscript with our format requirements, which are summarized on the following checklist:

Communications Psychology formatting checklist

and also in our style and formatting guide Communications Psychology formatting guide .

* **CODE AVAILABILITY:** All Communications Psychology manuscripts must include a section titled "Code Availability" at the end of the methods section. In the event of publication, we require that the custom analysis code supporting your conclusions is made available in a publicly accessible repository; at publication, we ask you to choose a repository that provides a DOI for the code; the link to the repository and the DOI will need to be included in the Code Availability statement. Publication as Supplementary Information will not suffice. We ask you to prepare code at this stage, to avoid delays later on in the process.

* **DATA AVAILABILITY:**

All Communications Psychology manuscripts must include a section titled "Data Availability" at the end of the Methods section or main text (if no Methods). More information on this policy, is available at <http://www.nature.com/authors/policies/data/data-availability-statements-data-citations.pdf>.

At a minimum the Data availability statement must explain how the data can be obtained and whether there are any restrictions on data sharing. Communications Psychology strongly endorses open sharing of data. If you do make your data openly available, please include in the statement:

We recommend submitting the data to discipline-specific, community-recognized repositories, where possible and a list of recommended repositories is provided at <http://www.nature.com/sdata/policies/repositories>.

If a community resource is unavailable, data can be submitted to generalist repositories such as figshare or Dryad Digital Repository. Please provide a unique identifier for the data (for example a DOI or a permanent URL) in the data availability statement, if possible. If the repository does not provide identifiers, we encourage authors to supply the search terms that will return the data. For data that have been obtained from publicly available sources, please provide a URL and the specific data product name in the data availability statement. Data with a DOI should be further cited in the methods reference section.

REVIEWERS' EXPERTISE:

Reviewer #1: Exercise science and cognition

Reviewer #2: Exercise science and cognition

REVIEWERS' COMMENTS:

Reviewer #1 (Remarks to the Author):

The authors investigated the acute effects of exercise on cognition in adults. The overall effect was small, with anecdotal evidence supporting the relation between exercise and cognition. Moderators were also examined, with exercise modality playing an important role. Currently, there are more than 20 meta-analyses on the acute effects, with several being conducted in the last 5 years. Thus, a meta-analysis needs to cover some novel aspect, as a few studies more than a previous meta-analysis is not sufficient. This work introduces a Bayesian approach, which has some advantages over other methods. However, this meta-analysis fails to consider confounders and participants' characteristics, has a flawed categorization of moderators, does not assess and control publication bias, and pools together studies with very different designs, although their eligibility criteria does not allow them. Thus, I cannot recommend to further consider this work, unless eligibility criteria are changed or studies are screened again and the analyses are conducted after using a theoretical framework for categorizing moderators.

TITLE

The title should already introduce the sample (young and middle-aged adults).

INTRODUCTION

The first paragraph needs to state why looking at acute effects of exercise is relevant. The effects are transient, but still might be meaningful to understand how our brain and body interacts. Additionally, acute effects may allow the preparation for situations that demand cognition.

The introduction lacks a rationale for investigating acute effects in adults only. I also wonder why only young and middle-aged adults were focused on. The authors need to make clear when they cite literature that refers to this age group.

In my opinion, the introduction is too long and references to the methods should be removed from this section. Moreover, the text on the advantage of the Bayesian approach should be reduced.

METHOD

The authors do not make restrictions on cognitive domains, but their keywords are focusing on specific domains. Search terms on the outcome level might be too narrow. To improve efficiency, the sample should be a keyword as well.

The eligibility criteria implies that only studies comparing an exercise with a control group were included. This leaves out the many cross-over studies that have been conducted. Additionally, did the authors include studies that had only post-test measures?

The study is lacking a justification for categorizing cognition in 7 domains. In most frameworks, motor skills are not a cognitive domain. Additionally, social cognition is missing. A justification for grouping exercise durations and post-exercise delay is also missing.

The effect size calculation section implies that also studies without a control group were included. In this case, the pre-post change was used. In studies with a control group, the change was contrasted against the change of the control group. These are entirely different methods that should not be pooled together. The same applies to studies using post-measures only. These different methods do not yield comparable results. Additionally, the way the eligibility criteria are formulated, these studies should not have been included.

From the methods section, it remain unclear how cognitive tests and outcomes were categorized into the cognitive domains. Additionally, the synthesis of the data for yielding an effect size is not transparent. For example, if there was a Flanker task, was accuracy and reaction time on incongruent trials pooled together or treated separately?

The extraction and categorization of moderators is not described in sufficient detail. How was exercise modality categorized and what was the theoretical framework for this?

Methods to assess and correct for publication bias are completely missing. Methodological moderators have not been considered despite the indications from previous research.

RESULTS

From reading the results, I received more information on what the authors actually did. Exercise type considered different forms, but actually, the categorization is flawed. Exercise types can be categorized on metabolic demands (e.g. aerobic, anaerobic), skill (open-skill, closed-skill), and so on. One categorization could have been resistance exercise, endurance exercise and coordinative exercise. However, HIIT, cycling, running and circuit could all be categorized into one domain. Sports activity is not informative as this could include all other categories. Consequently, this analysis is not helpful.

The same applies to task outcome. It does not make sense to pool accuracy and RT. On an easy task, accuracy is generally high and changes are only expected in RT. Let's say there is a 5% improvement in RT, but no change in accuracy due to a ceiling effect. A combined analyses would reduce the improvement to 2.5%, not knowing that no change in accuracy was expected anyway. Moreover, the primary outcome depends on the task. For working memory tasks, accuracy or hit rate is more important than for attention tasks.

It seems that no moderator analysis has been performed on methodological variables and participants' characteristics. However, we know that these are important and they should be controlled.

DISCUSSION

I cannot comment on the discussion with the current results, because the categorization of moderators is flawed, confounders have not been controlled for, and the RT and accuracy outcomes were pooled, although this should not be done.

Reviewer #2 (Remarks to the Author):

The authors provide a comprehensive meta-analytic review of the vast literature on acute exercise effects on cognitive function. There are several strengths to this study which make this paper a necessary contribution to the field. However, before this study should be accepted for publication in Comms Psychology, there are several important considerations for the authors to include in their discussion of the literature and in their statistical approach. These are outlined in more detail below [Editorial note: Reviewer #2 comments are included in a separate document]

Summary of Major Changes

We sincerely appreciate the time that both reviewers took to provide feedback on our manuscript. Both reviewers expressed concerns about the timeliness of our literature search, how moderators were categorized, including additional moderators that reflect participant and study characteristics (e.g., age, BMI, publication year), testing for the presence of publication bias, and our study inclusion criteria. To address these concerns, we made several significant changes that we summarize first before treating each reviewer's comments in turn.

First, R2 recommended that studies within the last three years should be included in the current meta-analysis. We agree with this recommendation, and expanded our search criteria to include articles published between the years 2020-2023 (Page 8, Line 161). The search yielded an additional 1,303 studies, bringing the total number of screened articles to 15,900. 21 of these additional articles were found to meet our search criteria, bringing the total number of studies included in the analysis to 113 which corresponded to 642 effect sizes. After including these more recent studies in our analysis, we observed moderate evidence in favor of exercise having a small positive effect on overall cognitive task performance (Page 15, Lines 316-318).

Second, R1 was concerned about the way in which exercise type, duration, intensity, and the time of exercise relative to task completion were categorized. When deciding on the levels of each moderator, we strove for a balance between statistical power and temporal resolution. For example, studies that tested cognition post-exercise did so either immediately after or more than 180 minutes after exercise cessation (see Page 17, Table 1). Those that tested cognition between these time points were binned together to ensure estimated regression weights were not spurious. The time bins of exercise duration were chosen such that each bin had a roughly equivalent number of effect sizes. Similarly, when categorizing exercise intensity, we both followed the American College of Sports and Medicine guidelines (Page 9, Lines 191-194) while also ensuring that each category contained an equivalent number of effects. Levels of exercise type were chosen based on what authors reported in each study, as stated on Page 9, Lines 197-199. Sport-related exercises (e.g., boxing, rock climbing, soccer) were aggregated into a single category given that few studies used this modality.

Third, both R1 and R2 expressed concern that a theoretical framework for how cognitive domains were categorized was missing. The chosen cognitive domain labels were generally based on the neurocognitive domains defined by the DSM-5 (Sachdev et al., 2014). The DSM-5 includes the following domains: executive function, perceptual-motor function, language, complex attention, social cognition, learning and memory. However, our literature search did not yield a sufficient number of studies for the estimation of robust effect sizes for some of these domains (e.g., language, emotion recognition), reflecting the fact that these domains are seldom studied in the acute exercise literature. Cognitive tasks were categorized into cognitive domains

based on author experience and domain assignment typically used in the literature (e.g., 2-back is working memory, Stroop effect is executive function, etc). To increase transparency in our analysis, we included our classification criteria as a supplemental table (Sheet: Cognitive Task Classification Criteria).

Fourth, both reviewers suggested that methodological moderators (R1) and participant characteristics (R2) be included in the analyses. We agree and included the following moderator analyses in our updated manuscript: publication year, within- vs between-subject design, and average sample age, BMI, VO2 max, height, weight, and the percentage of female subjects (Page 11 Lines 236-241).

Fifth, R1 raised concerns about publication bias. Prior to updating our literature search to include more recent studies, the Egger's regression test we conducted indicated that there was only anecdotal evidence for the intercept being different from zero and that our funnel plot was symmetric, suggesting that the presence of publication bias was low. After updating our literature search, though, there was strong evidence for a non-zero intercept term and asymmetry in our funnel plot. To control for this asymmetry, we applied the trim and fill method to interpolate effect sizes from small-sample studies that might bias our estimates (Duval and Tweedie, 2000).

Lastly, both R1 and R2 raised concerns about the eligibility criteria for studies that were included in the current analysis. More specifically, on why the current analysis was centered on acute effects in young adults between the ages of 18-45 years old. We chose to focus on this age range for two reasons. First, recent exercise research has predominantly been dedicated toward studying the effects in children and older adults (Erickson et al., 2019; Hillman et al., 2020; Liu et al., 2020; Ludyga et al., 2020; Raine et al., 2020; Sanders et al., 2019)(Page 6 Lines 115-117). Second, the vast majority of our own research program is dedicated toward understanding the interaction between cognition and exercise in healthy young adults and that is where our expertise lies. That, coupled with the fact that research funds supporting this work were focused on a particular adult group, is why we centered our analyses on this age range. Note, prior meta-analyses have used a more conservative age range (18-30) when classifying subjects as young adults (Chang et al., 2012; Moreau & Chou, 2019).

Reviewer #1

The authors investigated the acute effects of exercise on cognition in adults. The overall effect was small, with anecdotal evidence supporting the relation between exercise and cognition. Moderators were also examined, with exercise modality playing an important role. Currently, there are more than 20 meta-analyses on the acute effects, with several being conducted in the last 5 years. Thus, a meta-analysis needs to cover some novel aspect, as a few studies more than a previous meta-analysis is not sufficient. This work introduces a Bayesian approach, which has some advantages over other methods. However, this meta-analysis fails to consider confounders and participants' characteristics, has a flawed categorization of moderators, does not assess and control publication bias, and pools together studies with very different designs, although their eligibility criteria does not allow them. Thus, I cannot recommend to further consider this work, unless eligibility criteria are changed or studies are screened again and the analyses are conducted after using a theoretical framework for categorizing moderators.

TITLE

- The title should already introduce the sample (young and middle-aged adults).

INTRODUCTION

1. The first paragraph needs to state why looking at acute effects of exercise is relevant. The effects are transient, but still might be meaningful to understand how our brain and body interacts. Additionally, acute effects may allow the preparation for situations that demand cognition.

Response: First we thank the reviewer for all of their suggestions. Second, with respect to motivating acute exercise and cognition research, Page 3 Line 60 - Page 4 Line 63 was edited to include the following: "Studying the impact of single exercise sessions on cognition can provide insight into how changes in our body's physiological state impacts behavior. This understanding can then guide the creation of more effective longer-term exercise interventions, which essentially involve regularly repeating brief exercise sessions over an extended period."

2. The introduction lacks a rationale for investigating acute effects in adults only. I also wonder why only young and middle-aged adults were focused on. The authors need to make clear when they cite literature that refers to this age group.
 - Response: Please refer to the summary of major changes for how this was addressed in the updated manuscript.

3. In my opinion, the introduction is too long and references to the methods should be removed from this section. Moreover, the text on the advantage of the Bayesian approach should be reduced.
 - Response: The following sentence: “Lastly, Bayesian meta-analysis naturally integrates into decision making frameworks (Hackenberger, 2020; Sutton et al., 2001; Sutton & Abrams, 2001)” on Page 7, Lines 142-143 of the previous manuscript has been removed from the Introduction. However, we believe that it is important to introduce readers that may not be familiar with Bayesian statistics to its advantages when compared to the frequentist approach early on.

METHOD

4. The authors do not make restrictions on cognitive domains, but their keywords are focusing on specific domains. Search terms on the outcome level might be too narrow. To improve efficiency, the sample should be a keyword as well.
 - Response: Similar to previous meta-analyses (Chang et al., 2012; Lambourne & Tomporowski, 2010), our search string included cognitive keywords to yield studies investigating the relationship between exercise and cognitive task performance. This also reduces the number of returned search results by excluding studies investigating other factors, such as mood, making the screening process more feasible. The sample characteristics were included in the search using filters (see Page 8, Lines 161-168).
5. The eligibility criteria implies that only studies comparing an exercise with a control group were included. This leaves out the many cross-over studies that have been conducted. Additionally, did the authors include studies that had only post-test measures?
 - Response: As detailed on Page 8, Lines 170-174, we also included studies that implemented a pre/post exercise experimental design. The goal of the meta-analysis was to determine impact of exercise on cognition, thus requiring a reference group or condition that exercise induced effects could be compared to.
6. The study is lacking a justification for categorizing cognition in 7 domains. In most frameworks, motor skills are not a cognitive domain. Additionally, social cognition is missing. A justification for grouping exercise durations and post-exercise delay is also missing.
 - Response: Please see the summary of major changes for how this was addressed in the updated manuscript.

7. The effect size calculation section implies that also studies without a control group were included. In this case, the pre-post change was used. In studies with a control group, the change was contrasted against the change of the control group. These are entirely different methods that should not be pooled together. The same applies to studies using post-measures only. These different methods do not yield comparable results. Additionally, the way the eligibility criteria are formulated, these studies should not have been included.
 - Response: The intention of a meta-analysis is to understand the effect of a study within the context of other studies in the literature, and if this effect is consistent across a body of data (Borenstein et al., 2021). The calculated effect sizes across all of the designs included in the current meta-analysis represent a relative change in behavioral performance compared to a reference condition/group as a function of exercise. To test the influence of these designs on the estimated effect of exercise on cognitive task performance, we conducted a moderator analysis for experimental design (within vs between). The results are reported in Table 1 and Table 2. Lastly, as stated on Page 8, Lines 170-177, the eligibility criteria included all of these designs.
8. From the methods section, it remains unclear how cognitive tests and outcomes were categorized into the cognitive domains. Additionally, the synthesis of the data for yielding an effect size is not transparent. For example, if there was a Flanker task, was accuracy and reaction time on incongruent trials pooled together or treated separately?
 - Response: See summary of major changes.
9. The extraction and categorization of moderators is not described in sufficient detail. How was exercise modality categorized and what was the theoretical framework for this?
 - Response: See summary of major changes.
10. Methods to assess and correct for publication bias are completely missing. Methodological moderators have not been considered despite the indications from previous research.
 - Response: See summary of major changes.

RESULTS

11. From reading the results, I received more information on what the authors actually did. Exercise type considered different forms, but actually, the categorization is flawed. Exercise types can be categorized on metabolic demands (e.g. aerobic, anaerobic), skill (open-skill, closed-skill), and so on. One categorization could have been resistance exercise, endurance exercise and coordinative exercise. However, HIIT, cycling, running and circuit could all be categorized into one domain. Sports activity is not informative as this could include all other categories. Consequently, this analysis is not helpful.

- Response: We agree that there are multiple ways in which exercise can be categorized. However, what is arguably most informative for the field and designing future exercise protocols that are effective in eliciting changes in cognitive task performance is to know the specific type of exercise. As our analysis shows, different aerobic exercises can differently impact cognitive function. Categorizing them all into a single domain would obscure this relationship. In regard to classifying studies that implemented a sport activity (e.g., soccer, boxing, rock climbing, etc), a fine-grained categorization would yield too few studies/effects per class level to produce reliable estimates (see Table 1 for the number of total effects in the exercise type level Sport Activity). Page 9, Line 194- Page 10 201 was edited to state the following “Exercise types were based on the modality reported in each study, yielding the following categorizations: cycling, high intensity interval training (HIIT), running, walking, circuit training, resistance exercise, and sports activity. The latter category encompassed studies that used sports-related exercises that did not fit into the other labels, such as rock climbing or soccer.”

12. The same applies to task outcome. It does not make sense to pool accuracy and RT. On an easy task, accuracy is generally high and changes are only expected in RT. Let’s say there is a 5% improvement in RT, but no change in accuracy due to a ceiling effect. A combined analyses would reduce the improvement to 2.5%, not knowing that no change in accuracy was expected anyway. Moreover, the primary outcome depends on the task. For working memory tasks, accuracy or hit rate is more important than for attention tasks.

- Response: The goal of the meta-analysis was to determine the effect of acute exercise on general cognitive task performance. Subgroup analyses were then conducted to assess potential moderators of this relationship. Our subgroup analyses revealed that accuracy and RT are differentially impacted by exercise (Page 16, Lines 325-327; Figure 3A). We also tested for potential interactions between moderators, and observed that modulations in accuracy or RT are dependent on cognitive domain (Page 19 Line 370 - 372).

13. It seems that no moderator analysis has been performed on methodological variables and participants' characteristics. However, we know that these are important and they should be controlled.

- Response: See summary of major changes.

DISCUSSION

- I cannot comment on the discussion with the current results, because the categorization of moderators is flawed, confounders have not been controlled for, and the RT and accuracy outcomes were pooled, although this should not be done.

- Response: Each of these critiques have been addressed in our responses above.

Reviewer #2

The authors provide a comprehensive meta-analytic review of the vast literature on acute exercise effects on cognitive function. There are several strengths to this study which make this paper a necessary contribution to the field. However, before this study should be accepted for publication in *Comms Psychology*, there are several important considerations for the authors to include in their discussion of the literature and in their statistical approach.

GENERAL COMMENTS

- The authors provide a comprehensive meta-analytic review of the vast literature on acute exercise effects on cognitive function. There are several strengths to this study which make this paper a necessary contribution to the field. However, before this study should be accepted for publication in *Comms Psychology*, there are several important considerations for the author to include in their discussion of the literature and in their statistical approach. These are outlined in more detail below.
- Strengths:
 - Comparison between frequentist and Bayesian models; use of Bayesian models in the current paper ensures that this study is a strong contributor to the field.
 - Clear calculations for effect sizes for studies with pre/post, post, during, and comparison with control groups were well defined and justified.
 - 92 studies included in final analysis.
 - Interaction analysis between moderators is novel, interesting, and well-reported.

INTRODUCTION

1. Lines 45-50: The population of each of these studies differ in terms of age. This should be addressed during this review of the literature, as the effects of acute exercise on neurocognitive function differ vastly by age. E.g., Hillman (2009) was in preadolescent children; Kamijo (2004) was in young adults; Chen (2016) was in children; Schaefer (2010) both children and adults. Age is later mentioned in relation to Chang (2012) study on lines 72-75; however, this should be considered earlier when comparing the previously mentioned studies.

Response: First, we thank the reviewers for all of their excellent suggestions. Second, we agree that the impact of exercise on cognition can vary across different age groups, and have removed the following citations to reflect the scope of the meta-analysis on cognition in young adults: Hillman et al., 2009; Chen 2016.

2. A great deal of work went into characterizing the literature from 1995-2020. However, when this meta-analysis will be published (i.e., 2023 or 2024), there will be 3 years of unaccounted data. Thus, prior to publication, the authors do need to also include the most recently published articles (i.e., 2020-2023) on acute exercise and cognition to fully contribute to the field. A final database search (1995-2023) should be conducted, relevant papers should be added, and results should be re-analyzed to include these papers.

Response: As recommended, we expanded our search criteria to include articles published between the years 2020-2023 (Page 8, Line 161-). The search yielded an additional 1,303 studies, bringing the total number of screened articles to 15,900. 21 of these additional articles were found to meet our search criteria, bringing the total number of studies included in the analysis to 113 which corresponded to 642 effect sizes.

3. A large proportion of the acute literature is in child and adolescent populations, considering the importance of acute exercise during schooltime. The study described herein would strengthen drastically by including child and adolescent populations in their search, and not excluding them (i.e., non-clinical studies with subjects aged 5-45). This would also allow for further characterization of age in the Bayesian meta-analytic approach. Also, 45 seems like a young cutoff for adult studies. The authors should consider increasing their sample age to include all adults (i.e., up to 65). Otherwise, authors should comment on why adults aged 46-65 were not included, considering 65 is the typical cutoff period between adulthood and older adulthood. The authors comment on using a “wide net” to provide an updated and current state of the literature (i.e., by including both laboratory and real-world exercise), and as such, age should also be included in this approach.

Response: We appreciate the suggestion to include studies that tested cognition in children and adolescents, and agree that it would strengthen the current analysis. However, we unfortunately would not be able to screen the yielded studies from the expanded literature search within a manner timely for the review process. Replicating our original string with the constraint that subjects are between the ages of birth to 17 years old resulted in 6,186 results from the database PsycInfo. Similarly, expanding our search to include studies with sample sizes between the ages of 45-65 yielded more than 6,000 results. Note, prior meta-analyses have used a more conservative age range (18-30 years old) when classifying subjects as young adults (Chang et al., 2012; Moreau & Chou, 2019). Furthermore, the cut off between young adulthood and middle-age is typically between 40-45 years old.

4. Line 139: When describing the 4th advantage to using Bayesian statistics, the author should further clarify this point about integration into decision making frameworks. This

study will be among few Bayesian meta-analyses in the field, compared to the wealth of frequentist models. As such, the authors shouldn't assume all readers are familiar with Bayesian terminology.

Response: This statement has been removed in accordance with the other reviewers suggestions to reduce the amount of text introducing the bayesian approach. We intended to convey that the posterior distribution produced by the Bayesian approach could be used to make data-driven decisions when designing future exercise protocols. For instance, one could use the cumulative density function of the posterior distribution to determine the probability that a given exercise modality would elicit at least a small enhancement of cognitive task performance.

METHODS

5. Similar to the comment above regarding the search years (1995-2020), the cutoff date for accessing papers was January 2021, however, this paper has been sent for review in June 2023. We appreciate the authors have presumably worked hard on this paper since 2021, however, the current version of this study is missing important manuscripts published between January 2021 and June 2023 that are essential and would contribute to the novelty of the current study.

Response: See our comments to suggestion 2 about expanding the literature search to include these studies.

6. Line 177: Since executive functions are considered, working memory should also be considered as a separate domain. Does executive function include inhibitory control? If not, inhibitory control should also be included. I wonder if any outcomes are considered twice between the classification of attention and executive function?

Response: Working memory and inhibitory control were included as sub-domains of executive function (Page 21 Line 401-403). These were not analyzed separately considering that many of the recent systematic reviews and meta-analyses conducted have focused predominantly on this cognitive domain (Cantelon & Giles, 2021; Ishihara et al., 2021; Ludyga et al., 2016). Furthermore, each of the other cognitive domains could also be divided into their subdomains, but the number of effects within each subdomain may not lead to reliable estimates (e.g., Motor Skill and Perception). Outcomes were not considered twice between domain classifications.

7. Intensity and duration coding are sufficient, but the cognitive domain codes likely require further attention as described above

Response: See summary of major changes.

8. The authors do a good job of describing the important moderator analyses: cognitive domain, measure of cognitive test relative to exercise, task outcome measure, exercise intensity, duration, and type. However, they haven't linked "Table 2" to the methods text where the moderators are mentioned (e.g., line 181). This is an issue because the early results report on effects of HIIT and cycling, but the reader hasn't been informed about what other exercise types are considered for this moderator analysis. The information is all in Table 2, but should be linked to the text to help the reader.

Response: A description of the exercise types has been included in the methods on Page 9, Line 190-194 "Exercise types were based on the modality reported in each study, yielding the following categorizations: cycling, high intensity interval training (HIIT), running, walking, circuit training, resistance exercise, and sports activity. The latter category encompassed studies that used sports-related exercises that did not fit into the other labels, such as rock climbing or soccer."

9. Also on moderator analyses, the authors have not considered crucial variables which reflect the advancements in this field lately (i.e., including/controlling for the effects of individual differences, such as BMI/adiposity, fitness, gender, etc... on post-exercise cognitive outcomes). These moderator analyses should be included in the analysis where appropriate (most studies report BMI, fitness; all studies report age and gender). When the authors reanalyze this dataset to include children and adolescents, they should also control for (i.e., with moderator analysis) for age and pubertal development.

Response: See summary of major changes for the additional moderators included to test the influence of participant characteristics.

10. The authors need to include a complete table of the 92 studies in the metaanalysis.

Response: A complete table of the studies included in the analyses has been provided as a table in the Supplementary documents.

RESULTS

11. Figure 1 demonstrates ProQuest and Google Scholar were used in the identification process. However, the methods state that PsychInfo and Google Scholar were used. This should be corrected and clarified.

Response: Figure 1 has been edited to state PsycInfo instead of ProQuest.

DISCUSSION

12. Lines 427-429: “selective impact on cognition”. This discussion point would be heightened if the authors also included individual differences (i.e., those mentioned above) as moderator analyses. The current literature on acute exercise effects on cognition demonstrates high sensitivity to fitness and obesity, and as such, the analyses and the discussion would be strengthened if these moderators are included in the analysis. This could also contribute to the null findings with exercise intensity and time of cognitive assessment.

Response: We appreciate this suggestion and tested the moderators average sample BMI, height, and weight (Table 2).

13. Considering a large portion of this literature focuses on “post-exercise” effects, and the authors also included “during-exercise” effects on cognitive function, to which there were surprisingly null results, the authors should address this point with more clarity in the discussion.

Response: We have discussed how these findings fit within the broader literature (Page 28, Line 522-527).

REFERENCES

14. This meta-analysis contributes to the literature on acute exercise effects on cognitive functions. However, the introduction (i.e., lines 63-65; lines 83-85) and discussion could be strengthened by citing several recently published meta-analyses on very similar topics which the authors have missed (there have been published meta-analytic reports since 2012), where there is considerable overlap among exercise/trained athlete type (e.g., aerobic, HIIT, resistance, etc) and/or cognitive domains (e.g., attentional allocation, inhibition, working memory, etc) between these 4 citations (provided below in order of publishing year) and the current study:
 1. Wang, S., Yin, H., Wang, X., Jia, Y., Wang, C., Wang, L., & Chen, L. (2019). Efficacy of different types of exercises on global cognition in adults with mild cognitive impairment: a network meta analysis. *Aging clinical and experimental research*, 31, 1391-1400.
 2. Haverkamp, B. F., Wiersma, R., Vertessen, K., van Ewijk, H., Oosterlaan, J., & Hartman, E. (2020). Effects of physical activity interventions on cognitive

outcomes and academic performance in adolescents and young adults: A meta-analysis. *Journal of sports sciences*, 38(23), 2637-2660.

3. Ishihara, T., Drollette, E. S., Ludyga, S., Hillman, C. H., & Kamijo, K. (2021). The effects of acute aerobic exercise on executive function: A systematic review and meta-analysis of individual participant data. *Neuroscience & Biobehavioral Reviews*, 128, 258-269.
4. Logan, Nicole E., Donovan A. Henry, Charles H. Hillman, and Arthur F. Kramer. "Trained athletes and cognitive function: A systematic review and meta-analysis." *International Journal of Sport and Exercise Psychology* 21, no. 4 (2023): 725-749.

Response: We appreciate the suggestion and included the last 3 citations in the discussion.

References

- Borenstein, M., Hedges, L. V., Higgins, J. P. T., & Rothstein, H. R. (2021). *Introduction to Meta-Analysis*. 10.1002/9780470743386
- Cantelon, J. A., & Giles, G. E. (2021). A Review of Cognitive Changes During Acute Aerobic Exercise. *Frontiers in Psychology, 12*. <https://doi.org/10.3389/FPSYG.2021.653158>
- Chang, Y. K., Labban, J. D., Gapin, J. I., & Etnier, J. L. (2012). The effects of acute exercise on cognitive performance: A meta-analysis. *Brain Research, 1453*, 87–101. <https://doi.org/10.1016/j.brainres.2012.02.068>
- Erickson, K. I., Hillman, C., Stillman, C. M., Ballard, R. M., Bloodgood, B., Conroy, D. E., Macko, R., Marquez, D. X., Petruzzello, S. J., & Powell, K. E. (2019). Physical Activity, Cognition, and Brain Outcomes: A Review of the 2018 Physical Activity Guidelines. *Medicine and Science in Sports and Exercise, 51*(6), 1242–1251. <https://doi.org/10.1249/MSS.0000000000001936>
- Hillman, C. H., McDonald, K. M., & Logan, N. E. (2020). A Review of the Effects of Physical Activity on Cognition and Brain Health across Children and Adolescence. *Nestle Nutrition Institute Workshop Series, 95*, 116–126. <https://doi.org/10.1159/000511508>
- Ishihara, T., Drollette, E. S., Ludyga, S., Hillman, C. H., & Kamijo, K. (2021). The effects of acute aerobic exercise on executive function: A systematic review and meta-analysis of individual participant data. *Neuroscience & Biobehavioral Reviews, 128*, 258–269. <https://doi.org/10.1016/j.neubiorev.2021.06.026>
- Lambourne, K., & Tomporowski, P. (2010). The effect of exercise-induced arousal on cognitive task performance: A meta-regression analysis. *Brain Research, 1341*, 12–24. <https://doi.org/10.1016/j.brainres.2010.03.091>

- Liu, S., Yu, Q., Li, Z., Cunha, P. M., Zhang, Y., Kong, Z., Lin, W., Chen, S., & Cai, Y. (2020). Effects of Acute and Chronic Exercises on Executive Function in Children and Adolescents: A Systemic Review and Meta-Analysis. *Frontiers in Psychology, 11*, 554915. <https://doi.org/10.3389/fpsyg.2020.554915>
- Ludyga, S., Gerber, M., Brand, S., Holsboer-Trachsler, E., & Pühse, U. (2016). Acute effects of moderate aerobic exercise on specific aspects of executive function in different age and fitness groups: A meta-analysis. *Psychophysiology, 53*(11), 1611–1626. <https://doi.org/10.1111/PSYP.12736>
- Ludyga, S., Gerber, M., Pühse, U., Looser, V. N., & Kamijo, K. (2020). Systematic review and meta-analysis investigating moderators of long-term effects of exercise on cognition in healthy individuals. *Nature Human Behaviour, 1–10*. <https://doi.org/10.1038/s41562-020-0851-8>
- Moreau, D., & Chou, E. (2019). The Acute Effect of High-Intensity Exercise on Executive Function: A Meta-Analysis. *Perspectives on Psychological Science, 14*(5), 734–764. <https://doi.org/10.1177/1745691619850568>
- Raine, L. B., Kao, S.-C., Drollette, E. S., Pontifex, M. B., Pindus, D., Hunt, J., Kramer, A. F., & Hillman, C. H. (2020). The role of BMI on cognition following acute physical activity in preadolescent children. *Trends in Neuroscience and Education, 21*, 100143. <https://doi.org/10.1016/j.tine.2020.100143>
- Sanders, L. M. J., Hortobágyi, T., la Bastide-van Gemert, S., van der Zee, E. A., & van Heuvelen, M. J. G. (2019). Dose-response relationship between exercise and cognitive function in older adults with and without cognitive impairment: A systematic review and meta-analysis. *PloS One, 14*(1), e0210036. <https://doi.org/10.1371/journal.pone.0210036>

14th Feb 24

Dear Mr Garrett,

Thank you for submitting a revised version of your manuscript titled "Acute Physical Activity has Selective Effects on Cognition in Young Adults". After careful consideration and discussion with my colleagues, I am sorry to have to tell you that we need to ask for further revisions before we can send your manuscript back to the reviewers.

We take this unusual course of action is taken occasionally in order to avoid unproductive rounds of review that ultimately reduce the chances of the manuscript obtaining a fair and objective evaluation.

In particular, we ask you to:

-Submit the completed PRISMA checklist as part of your revisions (http://prisma-statement.org/documents/PRISMA_2020_checklist.pdf?AspxAutoDetectCookieSupport=1)

-Include references to all studies reviewed in the main reference list as opposed to in a sperate document

If you adequately respond to these requests, we would be happy to send your manuscript back to reviewers.

We hope to receive your revised version as soon as possible.

Please use the link below to submit a suitably revised manuscript and updated response to referees when they are ready.

[link redacted]

If you have any questions, please contact me.

Best regards,

Dr Antonia Eisenkoeck

Senior Editor

on behalf of

Daniel Quintana, PhD

Editorial Board Member

Communications Psychology

orcid.org/0000-0003-2876-0004

Communications Psychology is committed to improving transparency in authorship. As part of our efforts in this direction, we are now requesting that all authors identified as ‘corresponding author’ create and link their Open Researcher and Contributor Identifier (ORCID) with their account on the Manuscript Tracking System prior to acceptance. ORCID helps the scientific community achieve unambiguous attribution of all scholarly contributions. You can create and link your ORCID from the home page of the Manuscript Tracking System by clicking on ‘Modify my Springer Nature account’ and following the instructions in the link below. Please also inform all co-authors that they can add their ORCIDs to their accounts and that they must do so prior to acceptance.

24th Apr 24

Dear Mr Garrett,

Thank you for your patience during the peer-review process. Your manuscript titled "Acute Physical Activity has Selective Effects on Cognition in Young Adults" has now been seen by 2 reviewers, and I include their comments at the end of this message. They find your work of interest but raised some remaining points. We are interested in the possibility of publishing your study in Communications Psychology, but would like to consider your responses to these concerns and assess a revised manuscript before we make a final decision on publication.

We therefore invite you to revise and resubmit your manuscript, along with a point-by-point response to the reviewers. Please highlight all changes in the manuscript text file.

Editorially, we consider it important that you address the request for separate analyses of the between- vs. within-subject effects. In regard to the categorization of exercise types, if you choose not to preform additional analyses, you may leave this as is and include this concern as a limitation. The requested modifications to the discussion and conclusion should also be addressed.

I am attaching an Editorial Requests Table that details critical reporting requirements for the revised manuscript. Please attend to each item and ensure your manuscript is fully compliant. We are requesting that your manuscript aligns with these requirements as this facilitates the evaluation of your manuscript, reducing delays in re-review and potential future acceptance. If your revised manuscript is not aligned with these requests on major issues, such as those concerning statistics, it may be returned to you for further revisions without re-review. Additional information can be found in our style and formatting guide Communications Psychology formatting guide.

Please use the following link to submit your

- revised manuscript,
- point-by-point response to the referees' comments,
- cover letter (as a separate document),
- the Editorial Policy Checklist (see below),

- the Reporting Summary (see below), and
- the completed Editorial Request Table (attached):

[link redacted]

Best regards,

Daniel Quintana

Daniel Quintana, PhD

Editorial Board Member

Communications Psychology

orcid.org/0000-0003-2876-0004

REVIEWERS' EXPERTISE:

Reviewer #1: Exercise science and cognition

Reviewer #2: Exercise science and cognition

REVIEWER REPORTS:

Reviewer #1 (Remarks to the Author):

The authors did a great job addressing the initial criticism and I also acknowledge the transparency regarding the analyses. In my opinion, the paper is in a good state, except two points that still struggle me.

7. I'm not convinced that a moderator analysis contrasting within vs between-subjects designs serves to address this issue. The problem is that simple pre-post designs without a control group will not adjust for learning effects, so that there is the summary effect + the learning effect, leading to imprecision. Thus, the moderator analysis must be conducted on the different types of calculating effect sizes.

9. "Levels of exercise type were chosen based on what authors reported in each study" - In my opinion, this is not an appropriate way. It makes more sense to use a categorization scheme that is based on a theoretical framework. The sports category could be swimming, but swimming would also be aerobic exercise.

Reviewer #2 (Remarks to the Author):

Responses:

The authors have sufficiently revised their manuscript with the previous suggestions, responded to the reviewer's comments, and provided additional sources and analyses where necessary. Additional comments on the revised paper are described below:

Revised Manuscript:

The authors provide neural explanations for the differences in RT and ACC, but have not provided behavioral markers of performance differences. Given that this study assesses cognitive function in the behavioral domain, they should also provide an equally in-depth explanation of the behavioral differences alongside hypothesized neural differences, i.e., why RT changes may be more sensitive compared to ACC changes (outside of the ceiling effect explanation).

In the conclusion, the authors state there is “moderate evidence for an acute bout of exercise enhancing overall performance”. However, throughout the manuscript, they state “The current meta-analysis observed that acute exercise has a small positive influence on overall cognitive task performance.” Caution on the use of terminology is recommended in the conclusion to not overstate the magnitude of the effect size when it was small effect, and this should be corrected.

EDITORIAL POLICIES

We ask that you ensure your manuscript complies with our editorial policies and reporting requirements.

To that end, we require revised manuscripts to be accompanied by two completed items: a reporting summary that collects information on study design and procedure, and an editorial policy checklist that verifies compliance with all required editorial policies.

Nature Research Reporting Summary

Editorial Policy Checklist

All points on the policy checklist must be addressed. Your revised manuscript can only be sent back to the referees if these checklists are completed and uploaded with the revision.

Notes: If you have submitted a Stage 1 Registered Report, Review, Primer, Comment, or Perspective you do not need to submit these forms. If you have already submitted these forms, you may disregard this request.

Reviewer #1

1. Comment: "I'm not convinced that a moderator analysis contrasting within vs between-subjects designs serves to address this issue. The problem is that simple pre-post designs without a control group will not adjust for learning effects, so that there is the summary effect + the learning effect, leading to imprecision. Thus, the moderator analysis must be conducted on the different types of calculating effect sizes."

Response: We acknowledge the importance of this concern, so to test whether the estimated overall effect size was driven by a learning effect, a separate meta-analysis was conducted on effect sizes from studies that tested cognition before and after an instance of exercise. If a learning effect in the no-control group designs is driving the overall pattern we observed, then one would expect there would be strong evidence of an even larger effect of exercise in that group. Not only was the estimated pooled effect size for this subset of data nominally similar to estimates for the entire dataset, there was also anecdotal evidence in favor of the null hypothesis ($g = 0.15 \pm 0.06$; HDI = [0.04, 0.24]; BF = 0.09). Furthermore, we tested for a difference in the estimated effect size for pre-/post- designs that either included or did not include an experimental control condition. There was strong evidence in favor of the null hypothesis for no difference between pre-/post- designs that either included or did not include a control (BF = 0.12), and strong evidence against non-zero parameter estimates (w/ control: $g = 0.18 \pm 0.10$; HDI = [0.03, 0.33]; BF = 0.51; w/o control: $g = 0.11 \pm 0.13$; HDI = [-0.03, 0.26]; BF = 0.18). Taken together, the estimated overall effect size is likely not solely driven by studies that used a pre-/post- design, nor the presence of a learning effect. These results are reported on Page 25-26 Line 349-359.

2. Comment: "Levels of exercise type were chosen based on what authors reported in each study" - In my opinion, this is not an appropriate way. It makes more sense to use a categorization scheme that is based on a theoretical framework. The sports category could be swimming, but swimming would also be aerobic exercise."

Response: As mentioned in our previous response to this point, we acknowledge that there are many ways in which types of exercise can be categorized. However, simply partitioning them into aerobic versus anaerobic exercises fails to capture the degree of variability between exercises within each of these groups (e.g., differences between the effects of cycling vs running). Further, such a categorization scheme does not provide a detailed description of the relationship between exercise and cognition that could be used to guide the design of future exercise studies and aid in the decision of which exercise modality to utilize. Note, an exercise modality is not purely aerobic or anaerobic, but rather these physiological states are dependent on a variety of factors including exercise intensity, duration, and individual fitness level. For example, maximal aerobic fitness tests conducted using either cycling or running, conventionally treated as aerobic exercises, can induce an anaerobic state as evidence by ventilatory and lactate thresholds (Rogers et al., 2021). Lastly, we opted to use the exercise modalities reported in each study to accurately reflect the current state of the literature and the types of exercises

being predominantly used. We have stated the following in the discussion to acknowledge that a potential limitation in the current analysis is how exercise modality was categorized: Page 10, Line 641 – Line 646 “A potential limitation in the current meta-analysis is the categorization of exercise type using the activity reported in each study. An alternative approach is to categorize exercise based on the theoretical and physiological distinctions between aerobic and anaerobic exercise. We did not adopt this approach here because many activities used in the literature typically include aerobic and anaerobic components, and basing their classification on what authors reported provides insights into the exercise modalities that have been predominantly used in the literature.”.

Reviewer #2

1. Comment: “The authors provide neural explanations for the differences in RT and ACC, but have not provided behavioral markers of performance differences. Given that this study assesses cognitive function in the behavioral domain, they should also provide an equally in-depth explanation of the behavioral differences alongside hypothesized neural differences, i.e., why RT changes may be more sensitive compared to ACC changes (outside of the ceiling effect explanation).”

Response: We once again appreciate the reviewer’s suggestions. The ceiling effect account we provide is just one possible explanation for the differential effect of exercise on RT and accuracy. We suggest that this is actually a behavioral account bolstered by neuroscientific evidence. Nevertheless, we have expanded this section to include another possible behavioral account based on the known sensitivities of accuracy and RT. Specifically, we have added the following on Page 34, Line 493 – 506 “Lastly, the differential impact of exercise on accuracy and RT may be due to the relative sensitivities of these dependent measures to modulations of different stages of information processing. For example, there is evidence that in near-threshold tasks accuracy is sensitive to perceptual manipulations, whereas in supra-threshold (i.e., perceptually easy tasks, including many of those used in the studies in this meta-analysis) RT is sensitive to modulations in both perceptual and post-perceptual processes (Mordkoff & Egeth, 1993; Santee & Egeth, 1982). Indeed, Davranche et al., (2023) utilized a drift diffusion model to determine which aspects of decision-making are modulated by HIIT. Importantly, drift rate and decision response boundary size increased significantly after exercise relative to before, while non-decision time decreased. This suggests there was an improvement in perceptual discrimination, the efficiency of non-decisional processes (e.g., motor execution), and the adoption of a more conservative criterion. Future research employing computational models of response time and representational fidelity is needed to develop a comprehensive understanding of the selective influence exercise on information processing speed and accuracy.”

2. Comment: “In the conclusion, the authors state there is “moderate evidence for an acute bout of exercise enhancing overall performance”. However, throughout the manuscript, they state “The current meta-analysis observed that acute exercise has a small positive

influence on overall cognitive task performance.” Caution on the use of terminology is recommended in the conclusion to not overstate the magnitude of the effect size when it was small effect, and this should be corrected.”

Response: In this context “moderate” is being used to describe the amount of evidence whereas “small” is being used to describe the size of the effect. These are two different things. We certainly want to avoid confusion, so we have revised description of the effect in the concluding remarks to make it explicit that the magnitude of the estimated effect was small: Page 40 Line 641 “In summary, the current meta-analytic examination has shown that there is moderate evidence for an acute bout of aerobic exercise inducing a small enhancement in overall cognitive performance...”.

References

- Davranche, K., Dorian, G., Arnaud, H., & Thibault, G. (2023). *Exploring Decision-Making Processes: The Impact of Intense Exercise* (p. 2023.02.14.528466). bioRxiv. <https://doi.org/10.1101/2023.02.14.528466>
- Mordkoff, J. T., & Egeth, H. E. (1993). Response time and accuracy revisited: Converging support for the interactive race model. *Journal of Experimental Psychology: Human Perception and Performance*, 19(5), 981–991. <https://doi.org/10.1037/0096-1523.19.5.981>
- Rogers, B., Giles, D., Draper, N., Mourot, L., & Gronwald, T. (2021). Detection of the Anaerobic Threshold in Endurance Sports: Validation of a New Method Using Correlation Properties of Heart Rate Variability. *Journal of Functional Morphology and Kinesiology*, 6(2), Article 2. <https://doi.org/10.3390/jfmk6020038>
- Santee, J. L., & Egeth, H. E. (1982). Do reaction time and accuracy measure the same aspects of letter recognition? *Journal of Experimental Psychology. Human Perception and Performance*, 8(4), 489–501. <https://doi.org/10.1037//0096-1523.8.4.489>

6th Jun 24

Dear Mr Garrett,

Thank you for your patience during the peer-review process. Before we move ahead with your manuscript, we would like to clarify an inconsistency in your manuscript. On page 26 it is written that, "Despite the estimated pooled effect size for this subset of data being nominally similar to the estimate for the entire dataset, there was anecdotal evidence in favor of the null hypothesis ($g = 0.15 \pm 0.06$; HDI = [0.04, 0.24]; BF = 0.09). A BF of 0.09 would be considered strong evidence for the null hypothesis, relative to the alternative, not anecdotal evidence. This also does not align with what is reported in Table 3's moderator analysis of study type, which shows stronger evidence in favor of the exercise effect appearing in the within-person studies. If there is indeed strong evidence for the null hypothesis, then this would suggest the overall effect of acute exercise having a small beneficial effect on general cognition seems to disappear when only pre-/post-test design studies are included, which has implications for the overall paper conclusion.

I am attaching an Editorial Requests Table that details critical reporting requirements for the revised manuscript. Please attend to each item and ensure your manuscript is fully compliant. We are requesting that your manuscript aligns with these requirements as this facilitates the evaluation of your manuscript, reducing delays in re-review and potential future acceptance. If your revised manuscript is not aligned with these requests on major issues, such as those concerning statistics, it may be returned to you for further revisions without re-review. Additional information can be found in our style and formatting guide Communications Psychology formatting guide.

Please use the following link to submit your

- revised manuscript,
- point-by-point response to the referees' comments,
- cover letter (as a separate document),
- the Editorial Policy Checklist (see below),
- the Reporting Summary (see below), and
- the completed Editorial Request Table (attached):

[link redacted]

Best regards,

Daniel Quintana

Daniel Quintana, PhD

Editorial Board Member

Communications Psychology

orcid.org/0000-0003-2876-0004

EDITORIAL POLICIES

We ask that you ensure your manuscript complies with our editorial policies and reporting requirements.

To that end, we require revised manuscripts to be accompanied by two completed items: a reporting summary that collects information on study design and procedure, and an editorial policy checklist that verifies compliance with all required editorial policies.

Nature Research Reporting Summary

Editorial Policy Checklist

All points on the policy checklist must be addressed. Your revised manuscript can only be sent back to the referees if these checklists are completed and uploaded with the revision.

Notes: If you have submitted a Stage 1 Registered Report, Review, Primer, Comment, or Perspective you do not need to submit these forms. If you have already submitted these forms, you may disregard this request.

19th Jun 24

Dear Mr Garrett,

Your manuscript titled "Acute Physical Activity has Selective Effects on Cognition in Young Adults: A Meta-Analysis" has now been reviewed by our editorial team. I am delighted to say that we are happy, in principle, to publish a suitably revised version in *Communications Psychology* under the open access CC BY license (Creative Commons Attribution v4.0 International License).

We therefore invite you to revise your paper one last time to address a list of editorial requests. At the same time we ask that you edit your manuscript to comply with our format requirements and to maximise the accessibility and therefore the impact of your work.

EDITORIAL REQUESTS:

SUBMISSION INFORMATION:

OPEN ACCESS:

Communications Psychology is a fully open access journal. Articles are made freely accessible on publication under a CC BY license (Creative Commons Attribution 4.0 International License). This license allows maximum dissemination and re-use of open access materials and is preferred by many research funding bodies.

For further information about article processing charges, open access funding, and advice and support from Nature Research, please visit <https://www.nature.com/commspsychol/article-processing-charges>

At acceptance, you will be provided with instructions for completing this CC BY license on behalf of all authors. This grants us the necessary permissions to publish your paper. Additionally, you will be asked to declare that all required third party permissions have been obtained, and to provide billing information in order to pay the article-processing charge (APC).

* **DATA AVAILABILITY:**

[link redacted]

Best regards,

Jennifer Bellingtier

Jennifer Bellingtier, PhD

Senior Editor

Communications Psychology

Daniel Quintana, PhD

Editorial Board Member

Communications Psychology

orcid.org/0000-0003-2876-0004